# Bimetallic Lanthanum-Cerium-Loaded HZSM-5 Composite for Catalytic Deoxygenation of Microalgae-Hydrolyzed Oil into Green Hydrocarbon Fuels

**DOI:** 10.3390/molecules27228018

**Published:** 2022-11-18

**Authors:** Mustafa Jawad Nuhma, Hajar Alias, Muhammad Tahir, Ali A. Jazie

**Affiliations:** 1Department of Chemical Engineering, School of Chemical and Energy Engineering, Universiti Teknologi Malaysia (UTM), Johor Bahru 81310, Johor, Malaysia; 2Chemical Engineering Department, College of Engineering, University of Al-Qadisiyah, Al-Diwaniyah City P.O. Box 1767, Iraq; 3Chemical and Petroleum Engineering Department, United Arab Emirates University (UAEU), Al Ain P.O. Box 15551, United Arab Emirates

**Keywords:** *Chlorella vulgaris* microalgae, HZSM-5 zeolite, Lanthanum-Cerium, deoxygenation, hydrocarbons, oxygenates

## Abstract

Due to their high lipid content, microalgae are one of the most significant sources of green hydrocarbons, which might help lessen the world’s need for fossil fuels. Many zeolite-based catalysts are quickly deactivated by coke production and have a short lifetime. In this study, a bimetallic Lanthanum-Cerium (La-Ce)-modified HZSM-5 zeolite catalyst was synthesized through an impregnation method and was tested for the conversion of hydrolyzed oil into oxygen-free hydrocarbon fuels of high energy content. Initially, hydrolyzed oil (HO), the byproduct of the transesterification process, was obtained by the reaction of crude oil derived from Chlorella vulgaris microalgae and a methanol. Various catalysts were produced, screened, and evaluated for their ability to convert algal HO into hydrocarbons and other valuable compounds in a batch reactor. The performance of HZSM-5 was systematically tested in view of La-Ce loaded on conversion, yield, and selectivity. NH_3_-TPD analysis showed that the total acidity of the La-Ce-modified zeolites was lower than that of the pure HZSM-5 catalyst. TGA testing revealed that including the rare earth elements La and Ce in the HZSM-5 catalyst lowered the catalyst propensity for producing coke deposits. The acid sites necessary for algal HO conversion were improved by putting La and Ce into HZSM-5 zeolite at various loading percentages. The maximum hydrocarbon yield (42.963%), the highest HHV (34.362 MJ/Kg), and the highest DOD% (62.191%) were all achieved by the (7.5%La-2.5%Ce)/HZSM-5 catalyst, which was synthesized in this work. For comparison, the hydrocarbon yield for the parent HZSM-5 was 21.838%, the HHV was (33.230 MJ/Kg), and the DOD% was 44.235%. In conclusion, La and Ce-loading on the parent HZSM-5 may be responsible for the observed alterations in textural properties; nevertheless, there is no clear correlation between the physical features and the hydrocarbon yield (%). The principal effect of La and Ce modifying the parent HZSM-5 zeolite was to modify the acidic sites needed to enhance the conversion (%) of the algal HO during the catalytic deoxygenation process, which in turn raised the hydrocarbon yield (%) and increased the HHV and DOD%.

## 1. Introduction

The global research on alternative and sustainable energy sources has been sparked by the fact that fossil fuels are running out and that too much carbon dioxide is being released into the atmosphere [1,2,3,4,5,6]. Using petroleum derivatives made from fossil crude oil pollutes the environment by releasing greenhouse gases, which cause serious environmental problems related to global warming [7,8,9,10,11,12].

Many researchers have looked into how to make biofuels from both edible and non-edible biomass and how to grow them in different ways [13,14,15,16,17,18]. However, the biggest problem is that it takes up a lot of arable land and competes with food production around the world. Recently, microalgae have been suggested as some of the most promising options because they use photosynthesis, grow quickly, are easy to grow and, most importantly, do not take up space that could be used to grow food crops. Algae also produce about half of the world’s oxygen and use up a huge amount of CO_2_ [19].

Microalgae can grow in many different places around the world, including cold soil in Scandinavia and hot soil in the desert. They can also grow in both freshwater and saltwater. We can say that microalgae can grow anywhere on the surface of the planet. Up to 60% of microalgae are made up of triglycerides, which makes them a promising and important source of renewable energy. *Chlorella vulgaris* microalgae have a bright future in this field because they have a lot of fat in them [20].

Liquid biofuels have been developed with a lot of money and they may be the only alternative to traditional transportation fuels in the future [21]. However, because crude bio-oil still has a lot of oxygen atoms, a high freezing point, a high viscosity, a low heating value, and thermal instability, it cannot directly be used as a biofuel [22]. FAME (fatty acid methyl ester), which is made by transesterification, is the biodiesel that is most often used. The world’s supply of biodiesel went from 3.9 billion liters in 2005 to 18.1 billion liters in 2010 and it is expected to reach 41.4 billion liters in 2025 [23]. When it is mixed with fossil fuels in amounts of more than 20%, the engine needs to be changed. In addition, it is known that the high hygroscopicity of the biodiesel-diesel mixture, which is directly related to the presence of oxygenated molecules that help microorganisms grow during storage and cause unwanted solids to form, causes many problems, such as clogging filters. There is also the question of how much energy something has. Biodiesel has 33 MJ/L of energy, which is 9% less than regular diesel fuel [23]. All of these problems, which are mostly caused by FAME biodiesel’s high oxygen content, make it hard to frequently use [24].

Drop-in biofuels (bio-hydrocarbons) are an option because they can be used in any amount in current cars without having to change the engines or other mechanical parts. They can also be stored and moved using the same infrastructure. However, the cost and technical complexity are much higher than for biodiesel production through transesterification [13].

Hydrodeoxygenation (HDO) was developed as a different way to deal with the problems listed above. Several studies have looked at HDO at hydrogen pressures of 10–40 bar and temperatures of 260–350 °C, as well as different operating variables, such as reaction time, solvent use, and the ratio of catalyst to feed [13,14,15,16,17,18]. HDO is a type of hydrogenolysis that uses water to get rid of lipid oxygen molecules. In HDO, different kinds of catalysts are used. Even though HDO makes sure that pure hydrocarbons that can be used with regular fuels are made, the process is energy-intensive because it uses a high-pressure stream of hydrogen [24]. For 1 mol of reactant to be fully deoxygenated, at least 3–4 mol of hydrogen and a high hydrogen pressure of about 40 bar are required [25].

Catalytic deoxygenation is considered as an alternative to the HDO process. Decarboxylation, which produces CO_2_, and decarbonylation, which produces CO, both use up some of the carbon resources in the triglyceride feedstock. However, hydrodeoxygenation, which produces H_2_O, can turn most of the carbon resources in the feedstock into hydrocarbons [24]. Additionally, unlike hydrodeoxygenation, there is no water produced and the catalyst is not turned off [23].

Several studies have talked about catalytic deoxygenation using different reactants and catalysts with different operating parameters, such as reaction time, solvent use, and catalyst-to-feed ratio under different inert gases, such as N_2_ and Ar [18,26,27,28,29].

As a result, catalytic cracking has been one of the ways that vegetable oil has been turned into fuel for cars. During catalytic cracking, long chains of hydrocarbons are broken into smaller, lighter pieces. Oxygen molecules are taken out as CO_2_, CO, or H_2_O through decarboxylation and decarbonylation [30]. The catalyst makes the reaction happen at a lower temperature and makes more of the product [31]. Its properties, such as particle size, porosity, acidity, and surface area, affect the selectivity of the reaction pathway and the amount of product that is made [32].

Zeolite catalysts were the most common catalysts used to improve vegetable oil and bio-oil. Zhang et al. looked at the catalysts used in commercial operations to refine biofuel [33]. Even though the FCC catalyst and HZSM-5 were said to have the best performance for the catalytic cracking of bio-feedstocks, several zeolite-based catalysts had a short lifetime because coke formation deactivated them. Xu et al. showed that the acidic HZSM-5 has also been linked to the dealkylation, cracking, and aromatization processes that lead to the creation of coke. Even though the liquid had more hydrocarbons in the gasoline range, HZSM-5 made more gas products and needed less liquid [34]. In addition, most studies [35,36,37,38] have found that the HZSM-5 catalyst is the best zeolite-type catalyst for catalytic cracking to increase the number of hydrocarbons in bio-oils.

Most oxygenates can easily move to the active acid sites along the pore passage of HZSM-5 because of how well it is built [39]. Furthermore, the HZSM-5 catalyst has both Bronsted acid sites and Lewis acid sites, which are useful in acid-catalyzed processes. Bronsted acid sites are formed by thermal treatment and are linked to -OH groups. In contrast, Bronsted acid sites are more likely to be involved in catalytic deoxygenation reactions [39,40,41].

By dehydration, cracking, aromatization, isomerization, oligomerization, decarboxylation, and dealkylation, the presence of acid sites in the catalyst makes it easier for C-C and C-O bonds in the reactants to break and separate. Hydrocarbons and coke are also produced with the help of acid sites on catalysts [42]. Furthermore, the acidity of the catalyst should be just right to produce the best hydrocarbon yields while reducing the chance of more polymerization of hydrocarbons into coke materials. Coke is formed when heavy molecules are produced and stick to the surface of the catalyst, making it less effective as a catalyst [43]. During catalytic cracking, the active acid sites of the HZSM-5 structure are covered and the pores are blocked. This makes the hydrocarbons less active and less selective [43]. Low coke deposits may make the catalyst last longer but it is important to figure out how to stop coke from forming during the catalytic deoxygenation of the algal hydrolyzed oil of *Chlorella vulgaris* (HO). Higher hydrocarbon yields may depend on the acidity, porosity, and shape selectivity of the catalyst, as well as the size of the crystallites. All of these factors affect product yields and coke formation [35]. While this was going on, researchers looked into using metal-modified HZSM-5 zeolite to fine-tune the design and acidity of the catalyst. This led to better hydrocarbon selectivity, less coke formation, and a longer life for the catalyst.

In the catalytic upgrading of oxygenates, transition metals, such as nickel (Ni), molybdenum (Mo), zinc (Zn), and iron (Fe), are used as active components [41,43]. Even though transition metal-modified HZSM-5 catalysts do a great job of converting oxygenates to hydrocarbons, they are prone to coke formation [41]. Some transition metals, such as Platinum (Pt), Gold (Au), Ruthenium (Ru), Palladium (Pd), Rhodium (Rh), Silver (Ag), and Zirconium (Zr), based on the parent HZSM-5, are thought to be too expensive and not stable enough at high temperatures [39]. Several other researchers studied the catalytic upgrading of pyrolysis vapors using HZSM-5 zeolite frameworks filled with poor metals, such as Germanium (Ge), Gallium (Ga), Tin (Sn), and Aluminum (Al). They claimed that these cheap metals were not very stable at high temperatures and made more coke than HZSM-5 that had not been changed [39].

As of now, as far as we know, there is no study that looks at the process of catalytic deoxygenation for bio-oils or FAMEs for the composite Lanthanum-Cerium-modified HZSM-5 zeolite. Surprisingly, researchers want to put rare earth metals into HZSM-5 so that oxygenated pyrolysis gases can be upgraded into hydrocarbons with less coke production [44,45,46,47]. Sun et al., for example, looked at the ethylation of benzene with ethanol in a fixed-bed reactor with zeolites that had been changed with rare earth metals La and Ce. It was found that the number of strong Bronsted acid sites was slightly lower, which stopped coke from being made [48].

Strong acid sites are the main cause of coke deposition, which makes a catalyst stop working. Isha and Williams also show that the ability of cerium to store oxygen cuts down on coke production [44]. Researchers found that the rare earth metal particles were smaller, which let them go deeper into the zeolite channel and reduce the number of strong Bronsted acid sites. When a rare earth metal is used to impregnate HZSM-5, it makes the catalyst more stable at high temperatures [46,47]. However, the exchange of rare earth metal ions in the zeolite framework has helped the transfer of hydrogen atoms, which changes the production of hydrocarbons at the expense of oxygenated compounds. 

Zaki et al. [40], showed that the number of weak acid sites was deemed to be not catalytically important but it is assumed to effectively influence the proton mobility in zeolites.

According to the studies mentioned above, rare earth metals show promise in catalytic deoxygenation, which turns oxygenated molecules into hydrocarbons while lowering the amount of coke. Additionally, rare earth metals can prevent coking on the surface of the catalyst while also making it last longer. This makes it easier to turn oxygenated molecules into hydrocarbons. As far as we know, there has not been a lot of research on how La-Ce/HZSM-5 affects algal HO catalytic deoxygenation upgrading. Furthermore, most of the research on changing HZSM-5 with rare earth metals focused on model compound catalytic cracking processes using alcohol [39,40,41], methyl mercaptan [42,43], alkane [44], pyrolysis of biomass [45,46], and furans [47] instead of algal HO (FAMEs) as a reactant. 

Before the current investigation, we published two different studies that explored in detail the catalytic deoxygenation process for the algal HO under identical operating conditions (batch reactor, 300 °C, 1000 rpm, 7 bar of N_2_ inert gas (initial pressure), catalyst-to-algal HO ratio = 15% (wt.%), and 6 h). The first study focused on loading Lanthanum on the parent HZSM-5 zeolite at various loading weight percentages (5%La/HZSM-5, 10%La/HZSM-5, and 15%La/HZSM-5) [49], while the second focused on loading Cerium at various loading weight percentages (5%Ce/HZSM-5, 10%Ce/HZSM-5, and 15%Ce/HZSM-5) [50].

In this study, the composite Lanthanum-Cerium is looked at further as a bi-functional catalyst for the catalytic deoxygenation upgrading of algal HO that is individually loaded with different loading percentages on the parent HZSM-5. In this study, Lanthanum and Cerium were added to the parent HZSM-5 framework to turn oxygenates into hydrocarbons during catalytic deoxygenation and to keep coke formation to a minimum. 

The following four catalysts were made: HZSM-5, (2.5%La-7.5%Ce)/HZSM-5, (5%La-5%Ce)/HZSM-5, and (7.5%La-2.5%Ce)/HZSM-5. X-ray diffraction (XRD), nitrogen adsorption isotherms, scanning electron microscopy (SEM), temperature-programmed desorption of ammonia (NH_3_-TPD), and thermogravimetric analysis were used to characterize these materials. In a batch reactor, the performance of all of the synthesized catalysts for deoxygenating the algal hydrolyzed oil of *Chlorella vulgaris* microalgae was tested. It was discussed that the amount of Lanthanum-Cerium loading on the parent HZSM-5 affects the catalytic deoxygenation of algal HO to turn oxygenated compounds into hydrocarbons. The results were put into three groups: the product yield, the chemical composition, and the distribution of carbon numbers. Gas chromatography-mass spectrometry (GC-MS) was used to look at the chemicals in products. Extensive reaction paths were suggested for the catalytic deoxygenation of algal HO using all of the produced catalysts in this investigation. The properties of the liquid product were looked at, such as its elemental makeup, higher heating value (HHV), atomic ratios of O/C and H/C, and degree of deoxygenation (DOD%). All La-Ce-modified HZSM-5 with different loading weight percentages improved hydrocarbon yield (%), HHV, and DOD%. In this study, a (7.5%La-2.5%Ce)/HZSM-5 catalyst was synthesized, and its use led to the greatest possible hydrocarbon yield (42.963%), highest possible HHV (34.362 MJ/Kg), and highest possible DOD% (62.191%).

## 2. Experimental Section

### 2.1. Extraction of the Crude Oil from Chlorella vulgaris Microalgae

A Soxhlet extractor system was used to create the crude oil from *Chlorella vulgaris* microalgae. The original purpose of a Soxhlet extractor was to obtain a lipid out of a solid material. Most of the time, a soxhlet extraction is used when the target compound does not dissolve well in a solvent and the impurity does not dissolve in that solvent. Figure 1 shows how the Soxhlet extractor system works. It recirculates hexane by boiling and condensing it over and over again. Microalgae powder made from *Chlorella vulgaris* was put in the algae reservoir and 200 mL of liquid hexane was added to the hexane-oil reservoir. The hexane completely covers the powdered microalgae and breaks up a small amount of it. When a certain amount of hexane is in the algae reservoir, a siphon is made and the hexane and any oil it has dissolved drain into the hexane-oil reservoir. Here, a hot plate was used to heat the mixture of hexane and oil to (110 °C). The hexane is heated until it turns into a vapor, which rises through the tubes, shown by the dashed path in Figure 1. Oil, which has a higher boiling point, does not turn into vapor. When the vapor of hexane reaches the condenser tube, the hexane loses heat to the water that surrounds the tube. This causes the hexane to condense. The condensed hexane flows into the algae reservoir, where it soaks the microalgae powder and dissolves more oil. At 110 °C, the recirculation continues for eight hours. At the end of this extraction process, the hexane-oil reservoir is left with a mixture of hexane and oil, and the hexane-soaked filtration papers with leftover microalgae powder are left in the algae reservoir. At the end of the 8 h extraction time, the hexane-oil mixture was taken from the hexane-oil reservoir and kept well in the flask. It was then sent to the rotary evaporator to separate the hexane from the crude oil of *Chlorella vulgaris* microalgae. In the rotary evaporator machine, the mixture of hexane and oil is put into the hexane and oil reservoir and heated to 74 °C to remove the hexane an add it to another reservoir (the hexane reservoir), where the vaporized hexane passes through a condenser so that it can be recovered in the hexane reservoir [51]. In this study, the steps of the extraction process are shown in Figure 2.

### 2.2. Hydrolysis of the Crude Oil of Chlorella vulgaris Microalgae

A transesterification reaction was used to turn the crude oil of *Chlorella vulgaris* microalgae into fatty acid methyl esters (FAMEs), also known as algal hydrolyzed oil of *Chlorella vulgaris* microalgae (HO). The transesterification reaction was conducted with a hydrolysis setup that included a distillation column with a 250 mL round bottom flask with a stirrer. This flask was heated in a water bath, as shown in Figure 3. We weighed 5 g of the crude oil and heated it to 48 °C while stirring at 300 rpm for 10 min. At the same time, the methoxide (methanol-sodium hydroxide) solution was made by mixing 110 mL of methanol with 4.5 g of solid NaOH and shaking the mixture until all of the sodium hydroxide particles were dissolved. Due to the low lipid weight of the crude oil for *Chlorella vulgaris* microalgae, the amount of this solution (methoxide) was increased to be around 20 times greater than the crude oil weight. A total of 100 g of the methoxide solution was added to the heated crude oil, and this mixture was heated to 48 °C for 1 h while being stirred at 400 rpm [52]. After the flask cooled to room temperature, the mixture was neutralized with diluted sulfuric acid H_2_SO_4_ (1 M) to a pH of 7 and then it was put in the separating funnel for 10 h. In the separating funnel, there are two layers. The top layer is made up of FAMEs and the bottom layer is made up of glycerol. Figure 3 shows in detail how 5 g of the crude oil from the microalgae *Chlorella vulgaris* was turned into FAMEs.

### 2.3. Catalyst Preparation

To prepare the parent zeolite catalyst (HZSM-5) and the composite Lanthanum-Cerium-modified zeolite catalysts ((2.5%La-7.5%Ce)/HZSM-5, 5%La-5%Ce)/HZSM-5, and (7.5%La-2.5%Ce)/HZSM-5), first, ZSM-5 (NH_4_^+^) in the form of ammonium was turned into a proton (H^+^) form of zeolite (HZSM-5) through the calcination process. It was calcined at 600 °C for 4 h in static air (a ramp rate = 5 °C/min) [39]. To impregnate Lanthanum rare earth metal with Cerium rare earth metal on the support (HZSM-5) with different weight percentages, an incipient wetness technique and some changes to the method described by Dueso et al. were used [53]. For example, to make 10 g of (5%La-5%Ce)/HZSM-5 catalyst, 9 g of HZSM-5 was mixed with 250 mL of deionized water. The mixture was then mixed for 1 h at room temperature with 1.558 g of La(NO_3_)_2_.6H_2_O (Lanthanum nitrate hexahydrate) and 1.549 g of Ce(NO_3_)_2_.6H_2_O (Cerium nitrate hexahydrate).

The resulting slurry was heated to 90 °C while being stirred at a constant temperature until all the water evaporated and it became a paste. Next, the paste was dried in an oven at 110 °C for one night and then calcined at 750 °C for three hours to get rid of any impurities [54]. It was then put in a desiccator to cool to room temperature. The synthesized catalysts were then crushed with a mortar and pestle and packed well so they could be used later in this study. The remaining catalysts ((2.5%La-7.5%Ce)/HZSM-5 and (7.5%La-2.5%Ce)/HZSM-5) were created in the same manner.

### 2.4. Catalyst Characterization

Characterizations of synthesized catalysts were performed to learn more about their physical and chemical properties. All of the techniques for characterization were used in the standard way. X-ray diffraction was used to figure out how crystallized the catalysts were (XRD). Using a Philips (USA) PW1730 diffractometer at 40 kV and 30 mA with Cu as the anode material (k = 1.54 Å), we obtained powder XRD patterns. In the 2θ range of 5–100°, the scanning step size was 0.04°/min and each step took 1 s.

At −196 °C, a BELSORP-mini II (BEL Japan Inc., Toyonaka, Japan) was used to measure the isotherms for the adsorption of nitrogen. Before the samples were analyzed, they were degassed for 6 h at 200 °C under a vacuum to get rid of any compounds that had stuck to them. The Brunauer–Emmett–Teller (BET) and t-plot models were used to figure out the specific surface area and porosity. At p/p_0_ = 0.95, the total pore volume was worked out. The Barrett–Joyner–Halenda (BJH) method was used to figure out the surface area and size distribution of micropores and mesopores.

Scanning electron microscopy (MIRA III, TESCAN, Brno, Czech Republic) with an energy-dispersive X-ray spectroscopy (EDS) detector (SAMX, Trappes, France) was used to figure out the shape of the zeolite surface and the size of the crystals.

Using a NanoSORD-NS91 (Sensiran Co., Pardis, Iran) analyzer, the temperature-programmed desorption of ammonia (NH_3_-TPD) method was used to figure out how acidic the samples were. Thermogravimetric analysis (TGA) with a Q600 (TA, New Castle, DE, USA) in an atmosphere at a heating rate of 20 °C/min between 40 °C and 800 °C was used to measure the likelihood that coke would form on all newly made catalysts. This instrument was used in an air atmosphere to perform a thermal gravimetric analysis (TGA) of the parent zeolite HZSM-5 and Lanthanum-Cerium-modified HZSM-5 zeolite catalysts with different loading weight percentages. About 20 mg of the sample was put onto the instrument and kept there for 1 min at 40 °C. The sample was then heated to 720 °C at a rate of 20 °C per minute in dry air flowing at 100 mL/min. The experiments showed that the rate of change in the sample’s mass depends on the temperature (dm/dT, %/°C).

### 2.5. Catalytic Deoxygenation of the Algal HO Using the Parent HZSM-5, and Lanthanum-Cerium Modified Zeolites

Figure 4 shows that the algae (HO) was deoxygenated by a catalyst in a 100 mL stirring batch reactor (Zhengzhou Keda Machinery and Instrument Co., Zhengzhou, China) (ZZKD). A total of 23.6 g of algal HO and 3.54 g of the catalyst were mixed together and then poured into the reactor. To get rid of the air, the reactor was closed and compressed three times with five bars of nitrogen (N_2_). The initial pressure of N_2_ was then compressed to 7 bar and kept in the reactor. The reactor was then heated to 300 °C and maintained at that temperature for 6 h while the impeller rotated at 1000 rpm. When the reaction was completed, the mixtures were left to cool to room temperature. The gaseous state was produced (not studied). To separate the catalyst from the liquid phase, filtration was used. Figure 5 shows that a sample of the liquid product was taken and analyzed.

### 2.6. Product Analysis

Gas chromatography-mass spectrometry (GC–MS) was used to figure out what was in the algal HO and the liquid product. The gas chromatography system used was an Agilent Technologies 7820A GC System with a mass selective detector GC-5977E MSD in electron ionization (EI) mode at 70 eV. A 100% dimethylpolysiloxane Ultra Alloy Capillary Column UA-5MS column (P/N UA1-30M-1.OF, Frontier Laboratories Ltd., Koriyama, Japan) with an inner diameter of 250 μm, a film thickness of 0.25 μm, and a length of 30 m was used. The oven was kept at 45 °C for one minute before the temperature was raised by 6 °C per minute for 40 min until it reached 300 °C. The standard mass spectrum library from the National Institute of Standards and Testing (NIST) was used to figure out what the product chemicals were. The peak area percentage of the GC-MS chromatogram, which can also be called the yield percentage, can be used to figure out the relative fraction of product chemicals [55], which was shown to be Equation (1). Equation (2) [56] and the same method described by Katikaneni et al. were used to figure out the percentage of algal HO that was converted [57]. Equation (3) was used to figure out the average content (wt%) of X, where X = C, H, and O [58]. Equation (4) was used to figure out what the product’s higher heating value (HHV) was [58,59]. Equation (5) [60] was used to calculate the degree of deoxygenation (DOD%).
(1)$$\mathrm{Yield}\;\left(\%\right)=\frac{\mathrm{Area}\;\mathrm{of}\;\mathrm{the}\;\mathrm{desired}\;\mathrm{product}}{\mathrm{Area}\;\mathrm{of}\;\mathrm{all}\;\mathrm{detected}\;\mathrm{substances}}\times 100$$
(2)$$\mathrm{Conversion}\;\left(\%\right)=\frac{\mathrm{Mass}\;\mathrm{of}\;\mathrm{initial}\;\mathrm{compound}\;\mathrm{in}\;\mathrm{the}\;\mathrm{HO}-\mathrm{Mass}\;\mathrm{of}\;\mathrm{the}\;\mathrm{compound}\;\mathrm{in}\;\mathrm{the}\;\mathrm{product}}{\mathrm{Mass}\;\mathrm{of}\;\mathrm{initial}\;\mathrm{compound}\;\mathrm{in}\;\mathrm{the}\;\mathrm{HO}}\times 100$$
(3)$$\mathrm{wt}.\%\;X\;=\frac{\mathrm{Mass}\;\mathrm{of}\;X\;\mathrm{in}\;\mathrm{products}}{\mathrm{Mass}\;\mathrm{of}\;\mathrm{products}}$$
(4)$$\mathrm{HHV}\;\left(\frac{\mathrm{MJ}}{\mathrm{Kg}}\right)=-1.3675+0.3137\;\left(C\right)+0.7009\;\left(H\right)+0.0318\;\left(O\right)$$
(5)$$\mathrm{DOD}\%=\frac{\mathrm{Molar}\left(\frac{O}{C}\right)\mathrm{of}\;\mathrm{the}\;\mathrm{algal}\;\mathrm{HO}\;\mathrm{in}\;\mathrm{feed}-\mathrm{Molar}\;\left(\frac{O}{C}\right)\mathrm{of}\;\mathrm{the}\;\mathrm{catalytic}\;\mathrm{cracking}\;\mathrm{products}}{\mathrm{Molar}\left(\frac{O}{C}\right)\mathrm{of}\;\mathrm{the}\;\mathrm{algal}\;\mathrm{HO}\;\mathrm{in}\;\mathrm{feed}}$$

## 3. Results and Discussion

### 3.1. Catalyst Characterization

#### 3.1.1. XRD Results

Figure 6 and Table 1 show that X-ray diffraction (XRD) was used to check the purity and crystallinity of the parent HZSM-5, (2.5%La-7.5%Ce)/HZSM-5, (5%La-5%Ce)/HZSM-5, and (7.5%La-2.5%Ce)/HZSM-5 catalysts. The characteristic XRD peaks of HZSM-5 at 2θ = 7.96, 8.52, 14.8, 22.88, 24.24, 29.92, and 45.48° match the ZSM-5 peaks of the reference standard for a highly pure calcined ZSM-5. This means that the parent HZSM-5 zeolite samples have a typical MFI-type structure (mordenite framework inverted) [61,62,63,64,65]. The different X-ray sources used were thought to be the cause of the small shift in the characteristic angles from the reference standard peaks [66].

The XRD patterns of (2.5%La-7.5%Ce)/HZSM-5, (5%La-5%Ce)/HZSM-5, and (7.5%La-2.5%Ce)/HZSM-5 are very similar to those of the parent HZSM-5 in peak positions and shapes. This also shows that the Lanthanum-Cerium-modified HZSM-5 zeolites have a typical MFI (Mobil fifth) structure [67]. As a result, no impurity peaks were found in the XRD data and crystalline phases of Lanthanum and Cerium oxides were not found in the XRD patterns. This could mean that Lanthanum and Cerium oxides are either highly dispersed and amorphous on the outside of HZSM-5 or they are much smaller and have reached the channels of HZSM-5 [39,68]. As a result, using the incipient wetness impregnation method, the mentioned weight percentages of Lanthanum-Cerium did not change the parent HZSM-5 framework. This shows that all four of the synthesized zeolites have an MFI topology and are very pure.

The XRD patterns for (2.5%La-7.5%Ce)/HZSM-5, (5%La-5%Ce)/HZSM-5, and (7.5%La-2.5%Ce)/HZSM-5 show that the peak intensities are lower than in the parent HZSM-5, which means that the crystallinity is lower. This could be because there are more Lanthanum-Cerium species on HZSM-5 or because the crystallinity decreases as the Lanthanum-Cerium loading increases [64]. In general, the XRD intensities in the HZSM-5 pattern can tell if any species are in the channels [69]. Furthermore, the decrease in peak intensities in the patterns of (2.5%La-7.5%Ce)/HZSM-5, (5%La-5%Ce)/HZSM-5, and (7.5%La-2.5%Ce)/HZSM-5 shows that Lanthanum and Cerium species reach the channels of HZSM-5 [67,68,70].

The relative crystallinity of the (2.5%La-7.5%Ce)/HZSM-5, (5%La-5%Ce)/HZSM-5, and (7.5%La-2.5%Ce)/HZSM-5 was measured by the area of the peak between 2θ = 22.5° and 25° [71,72]. The crystallinity of the parent HZSM-5 catalyst was used as a reference and the results are shown in Table 1. However, Table 1 demonstrates that the crystallinity of all composite catalysts synthesized in this investigation is the same.

#### 3.1.2. Surface Analysis

Table 2 displays the BET surface area, micropore area, external surface area, and pore volume of the parent HZSM-5 and Lanthanum-Cerium-modified HZSM-5 zeolite catalysts with varying Lanthanum and Cerium loading weight percentages ((2.5%La-7.5%Ce)/HZSM-5, (5%La-5%Ce)/HZSM-5, and (7.5%La-2.5%Ce)/HZSM-5). It is clear in Table 2 that increasing the amount of Lanthanum and Cerium in the parent HZSM-5 catalyst had a big effect on all of these textural properties. The BET surface area of the parent HZSM-5 decreased from 338 m^2^/g to 272, 272, and 265 m^2^/g for (2.5%La-7.5%Ce)/HZSM-5, (5%La-5%Ce)/HZSM-5, and (7.5%La-2.5%Ce)/HZSM-5, respectively. In the same way, the HZSM-5’s micropore area decreased from 195 m^2^/g to 149, 161, and 171 m^2^/g, respectively. The external surface area of the parent HZSM-5 was also cut down, from 143 m^2^/g to 122, 111, and 94 m^2^/g, respectively. The total pore volume for HZSM-5 decreased from 0.223 cm^3^/g to 0.190, 0.189, and 0.180 cm^3^/g for (2.5%La-7.5%Ce)/HZSM-5, (5%La-5%Ce)/HZSM-5, and (7.5%La-2.5%Ce)/HZSM-5, respectively. When comparing the parent HZSM-5 to the Lanthanum-Cerium-modified HZSM-5, all of these textural properties are lower in the Lanthanum-Cerium-modified HZSM-5. This might be because Lanthanum and Cerium oxides build up on the mouth of the pores in the Lanthanum-Cerium-modified zeolite catalysts. Lanthanum and Cerium cations are smaller, so they can easily move through the pore mouths of HZSM-5 and settle in the internal pore channel of the parent HZSM-5 catalyst. This causes the values of all these textural properties to drop by a large amount [39,48,73,74]. It can also confirm the idea that the XRD intensities become weaker, as shown in the XRD pattern (see Figure 6) [63]. As shown in Table 2, the average particle size of the parent HZSM-5 increased after Lanthanum and Cerium were added to it. It increased from 17 nm to 22 nm, 21 nm, and 22 nm for (2.5%La-7.5%Ce)/HZSM-5, (5%La-5%Ce)/HZSM-5, and (7.5%La-2.5%Ce)/HZSM-5, respectively. The Lanthanum and Cerium loading covered some of the crystal’s surface and it looks like the loaded Lanthanum and Cerium was deposited on the crystal’s surface, causing the average particle size of the Lanthanum-doped catalysts to grow [75].

Figure 7 shows that, based on the IUPAC classification, the N_2_ adsorption-desorption isotherms of the Lanthanum-Cerium-modified zeolite catalysts are type I isotherms with an H_3_ hysteresis loop at a high P/P_0_ region. This is typical of micropore and mesoporous samples [63,76], which are very similar to those of the HZSM-5. However, when Lanthanum and Cerium were added to the original HZSM-5 catalyst, the N_2_ adsorption decreased as the relative pressure (P/P_0_) increased. The sharp rise in the amount of N_2_ that was absorbed by the samples in the low and medium pressure regions P/P_0_ (0–0.35) proved that a microporous zeolite material had formed [77].

The hysteresis loops come from the change in the uptake of nitrogen (N_2_) at a relative pressure (P/P_0_) in the range of 0.35–0.9, which shows the presence of slit-shaped pores [78,79]. The hysteresis loops come from the capillary condensation within the mesopores through nitrogen multilayers adsorbing on the inner surface [80]. As a result, both HZSM-5 and Lanthanum-Cerium-modified catalysts show a small step of N_2_ uptake at a relative pressure (P/P_0_) between 0.9 and 1.0. This is a sign of interparticle macroporosity [81]. Because of this, both the parent HZSM-5 and Lanthanum-Cerium-modified catalysts have both micropores and mesopores. 

#### 3.1.3. Ammonia TPD Analysis

NH_3_-TPD has been used to describe the strong acid sites and the weak acid sites on the parent HZSM-5 and Lanthanum-Cerium--modified HZSM-5 zeolite catalysts ((2.5%La-7.5%Ce)/HZSM-5, (5%La-5%Ce)/HZSM-5, and (7.5%La-2.5%Ce)/HZSM-5) [82,83]. Figure 8 shows the NH_3_-TPD profiles for the parent HZSM-5 and the Lanthanum-Cerium-modified HZSM-5 zeolite catalysts. The typical NH_3_-TPD profile for the parent HZSM-5 catalyst has two maximum peaks at low and high temperatures. The low-temperature desorption peak at 216 °C has been linked to the release of NH_3_ from Lewis sites (weak acid sites), such as extra-framework aluminum [84,85], while the high-temperature desorption peak at 439 °C has been linked to the release of NH_3_ from the Bronsted acid sites (strong acid sites) coming from framework aluminum [86,87,88]. As shown in Figure 8, the peak areas indicate the number of acid sites and the peak temperature show the acid strength [81]. Figure 8 also shows that, for all the Lanthanum-Cerium-modified HZSM-5 zeolite catalysts compared to the parent HZSM-5, when Lanthanum and Cerium are loaded (wt.%) on the parent HZSM-5, the number and strength of all of the weak and strong acid sites decrease to a different degree, which lowers the total acidity. The alteration of zeolites with metals alters the overall acidity of the parent zeolite, which is consistent with the findings of earlier studies [63,89,90].

In fact, the Lanthanum and Cerium species could enter the parent HZSM-5 zeolite tunnel by dispersing or exchanging with H^+^. This would change some of the strong Bronsted acid (B) centers into Lewis acid (L) centers, changing the surface acidity of the catalysts [91,92,93,94]. Moreover, Lewis-type acid sites in zeolite catalysts are mostly linked to aluminum species that are not part of the framework (extra-framework) [95], while aluminum in the zeolitic framework can cause strong acid sites [96]. Therefore, the fact that the number of Lewis acid sites is more than the number of Bronsted acid sites in doped catalysts suggests that there are more aluminum species that are not part of the framework. The XRD results suggest that the increase in aluminum species that are not part of the framework could have come from dealumination in the framework of HZSM-5. Therefore, the removal of the aluminum species from the framework can lead to a drop in the number of strong acid sites. Ouyang et al. showed that it would be helpful to stop coke deposits from forming if the number of strong acid sites were to decrease. It was also shown that reducing the number of strong acid sites made the active sites of the catalyst more stable [97].

As shown in Figure 8, for all of the Lanthanum-Cerium-modified zeolite catalysts, the low-temperature peak slightly moved to lower temperatures, with a peak profile that was similar to the parent HZSM-5 peak temperatures (at 216 °C), while the high-temperature desorption peak at (439 °C) clearly decreased, and in the case of (7.5%La-2.5%Ce)/HZSM-5, this peak disappeared. This is in line with the results of previous researchers who explained that the modification of zeolites with metals could lead to the disappearance of the high-temperature desorption peak and stated that the reason for the disappearance of this peak is due to the presence of metal ions and metal oxides. Consequently, the number and strength of sites are expected to depend on the content and location (e.g., specific cation exchange sites and extra framework) of the doped metal species [90,98].

The calibration curves of the TCD values were used to obtain the values of the desorbed NH_3_ in mmol/g, which are provided in Table 3. These values were acquired from the TCD values in volts (V). However, after Lanthanum and Cerium loaded on the parent HZSM-5, the total acidities of the Lanthanum-Cerium-modified zeolite catalysts were reduced from 0.740 mmol/g for the parent HZSM-5 to 0.524 mmol/g for (2.5%La-7.5%Ce)/HZSM-5, indicating that the loading of Lanthanum-Cerium affects the acidic characteristics of the parent HZSM-5 catalysts, which led to the decrease in both strong and weak acid site amounts. 

In conclusion, adding Lanthanum-Cerium to the parent HZSM-5 zeolite has a great effect on the total acid sites, especially the acidic Bronsted sites. However, this area of strong acid sites (the Bronsted sites) is thought to be the main catalytic center and include the dominant acid sites during catalytic deoxygenation reactions [32], whereas the number of weak acid sites (the Lewis sites) was deemed to be not catalytically important but it is assumed to effectively influence the proton mobility in zeolites [40]. 

#### 3.1.4. Thermogravimetric Analysis

Thermogravimetric analysis (TGA) was utilized to estimate the quantity of carbon that would fill the pores of freshly synthesized catalysts employed in the catalytic deoxygenation process in this study [99]. Figure 9 shows the results of the thermogravimetric analysis (TGA) for all of the freshly synthesized catalysts. It can be seen that the total mass loss for HZSM-5 is 5.4%, while it is 0.82%, 3.0%, and 3.8% for (2.5%La-7.5%Ce)/HZSM-5, (5%La-5%Ce)/HZSM-5, and (7.5%La-2.5%Ce)/HZSM-5, respectively, which is apparently due to the release of water from narrow channels [100,101], especially in the range of 30–170 °C, which was accompanied by water removal from the fresh samples [102]. As a result, adding Lanthanum and Cerium to the HZSM-5 catalyst makes it less likely that coke deposits form compared to the original HZSM-5. The (2.5%La-7.5%Ce)/HZSM-5 had the least mass loss.

#### 3.1.5. SEM Analysis

Figure 10 shows the surface structure of the parent HZSM-5 and Lanthanum-Cerium-modified HZSM-5 zeolite catalysts with different amounts of Lanthanum and Cerium. These are (2.5%La-7.5%Ce)/HZSM-5, (5%La-5%Ce)/HZSM-5, and (7.5%La-2.5%Ce)/HZSM-5. Figure 10 shows that all of the samples have nanometer-sized particles that are distinctly crystalline and grouped together. Images also show that some of the particles are broken up into smaller pieces during desilication [103,104]. These pieces can also be seen in pictures of Lanthanum-Cerium-modified zeolites, and adding rare earth metals does not seem to change the structure of the zeolites very much [63]. Both the pure HZSM-5 zeolite catalysts and the Lanthanum-Cerium-modified HZSM-5 zeolite catalysts have the same shape and structure of the crystallites. In addition, the surface of Lanthanum-Cerium-modified zeolite catalysts clumped together, which may have been caused by the connection of small particles during the calcination process. After the Lanthanum-Cerium metals were added, some small particles covered the surface, and these doped samples looked a little rougher than the pure HZSM-5 [39,63]. However, at this level of magnification, it was not possible to see how the Lanthanum-Cerium particles were spread out, which is in line with the XRD patterns. The average size of the crystallites in HZSM-5 and HZSM-5 modified with Lanthanum and Cerium is 100–400 nm.

### 3.2. Catalytic Deoxygenation of the HO for the Parent HZSM-5, (2.5%La-7.5%Ce)/HZSM-5, (5%La-5%Ce)/HZSM-5, and (7.5%La-2.5%Ce)/HZSM-5 Catalysts

In this study, all of the experiments were performed in the batch reactor under the same operating conditions (temperature, time, initial nitrogen pressure, catalyst-to-HO ratio (wt.), and stirring), which were 300 °C, 6 h, 7 bar of N_2_ inert gas (initial pressure), 15%, and 1000 rpm, respectively. This made it possible to compare the effects of the catalysts on conversion and product composition. The background for this study on catalytic deoxygenation, the operating conditions, were chosen based on studies completed with different types of reactants [13,43,105,106,107].

#### 3.2.1. Conversion of the Algal HO

Figure 11 and Table 4 show that the parent HZSM-5 catalyst had the lowest conversion of algal HO, 94.589%, of all the liquid products of the reactions conducted in this study. This could be because the HZSM-5 catalyst was not as effective under these reaction conditions. The percentage of algal HO conversion was higher in liquid products produced with Lanthanum-Cerium-modified zeolite than in liquid products produced with the parent HZSM-5. For the liquid product of catalytic deoxygenation for (2.5%La-7.5%Ce)/HZSM-5, (5%La-5%Ce)/HZSM-5, and (7.5%La-2.5%Ce)/HZSM-5, the algal HO conversion rates were 100%, 100%, and 98.035%, respectively. As a result, adding different amounts of Lanthanum-Cerium to HZSM-5 zeolite improved the acid sites that were needed to enhance the algal HO conversion.

The reason for this might be backed up by what other studies have found. As of now, as far as we know, there is no other study in the literature that is similar to the catalytic deoxygenation of algal HO performed in this study using the catalysts that were used. Therefore, this study’s operating conditions (such as the type of reactor, temperature, time, fatty acids or FAMEs as reactants, and the initial pressure of the pumped gas) were compared with those of close catalytic studies.

Tonya et al. looked at the conversion of soybean oil into hydrocarbons by catalytic deoxygenation for Pt/C, Pd/C, and Ni/C under an inert gas of nitrogen. They found that the conversion of soybean oil into hydrocarbons under the same operating conditions depends on the type of catalyst. The conversion percentages were 23%, 30%, and 93% for Pt/C, Pd/C, and Ni/C, respectively [105].

Mathias et al. studied the conversion of stearic acid for twenty different catalysts during catalytic deoxygenation at 300 °C and 6 bar of helium. They found that the percentage of stearic acid conversion depends on the type of catalyst, with 100% conversion for Pd/C and 4.6% conversion for Ir/SiO_2_ being the highest and lowest, respectively [106].

Botas et al. also showed that the conversion of rapeseed oil in the catalytic cracking of HZSM-5, Ni/HZSM-5, and Mo/HZSM-5 under the same operating conditions produces different conversion percentages for rapeseed oil with different amounts of compounds [43].

In conclusion, when Lanthanum-Cerium is added to nanocrystalline HZSM-5 zeolite, it changes its acid sites and textural properties. The Lanthanum-Cerium oxides are spread out across the zeolite support, with most of them staying in the micropores. Changes in the properties of the catalyst caused by adding metals also have a great effect on how well the catalyst works.

#### 3.2.2. Chemical Composition Group

In this study, the components and amounts of algal HO and the products of the catalytic deoxygenation of algal HO for the synthesized catalysts were determined using gas chromatography-mass spectrometry (GC-MS) and measured using the area normalization method. These results are shown in Figure 12a and Table 5. Algal HO contains 11.538% alkane, 61.12% esters, and 21.62% alcohol (phytol) by weight (%). Compared to the GC-MS results of algal HO and the liquid products of catalytic deoxygenation (see Table 5), the yield of phytol increased in all of the experiments performed in this study, while the yields of alkanes and esters decreased and were changed into other compounds (oxygenated compounds and non-oxygenated compounds). In conclusion, the phytol was not a reactant compound but it can be thought of as a product compound in this study. The alkanes with the fatty acid methyl esters can be thought of as reactant compounds.

In this study, the parts of the products were broken down into seven groups of bio-based chemicals. Two groups of non-oxygenated compounds (alkanes and alkenes) and five groups of oxygenated compounds (esters, ethers, aldehydes, ketones, and alcohols) made up the seven groups. Figure 12a and Table 5 show the composition groups of the algal HO and the liquid products of the catalytic deoxygenation of the algal HO in a batch reactor at 300 °C, for 6 h, at a 15% catalyst-to-HO weight ratio, with an initial pressure of 7 bar N_2_, and at 1000 rpm.

Taking into account the liquid products of catalytic deoxygenation of algal HO for the parent HZSM-5 zeolite catalyst, the total amount of oxygenated compounds (ester, ether, aldehyde, ketone, and alcohol) was higher than with the other Lanthanum-Cerium-modified zeolite catalysts, which were 12.64%, 4.93%, 6.471%, 2.394%, and 49.549%, respectively. Notably, when the products of the parent HZSM-5 catalyst were compared to those of the other Lanthanum-Cerium-modified zeolite catalysts, (2.5%La-7.5%Ce)/HZSM-5, (5%La-5%Ce)/HZSM-5, and (7.5%La-2.5%Ce)/HZSM-5, the parent HZSM-5 catalyst produced the highest percentage of esters (12.64%) and aldehydes (6.471%), the least amount of non-oxygenated compounds (hydrocarbons) was 21.833%, and those were made up of 4.785% alkane and 17.048% alkene. 

In this study, the highest amounts of alcohol and ketones were found in the liquid products of the catalytic deoxygenation of algal HO for the (2.5%La-7.5%Ce)/HZSM-5 zeolite catalyst. The percentages of alcohol and ketones in the liquid products were 50.5% and 5.861%, respectively. The produced proportions of the oxygenated compounds ester, ether, and aldehyde were 8.152%, 4.613%, and 1.52%, respectively. Most of this catalyst’s products were from the alcohol group. The produced amount of non-oxygenated compounds was 24.5% and those compounds were split between alkane (6.198%) and alkene (18.302%).

The second Lanthanum-Cerium-modified zeolite catalyst ((5%La-5%Ce)/HZSM-5) produced 35.149% of hydrocarbons, with 4.312% being alkane and 30.837% being alkene. On the other hand, for this catalyst ((5%La-5%Ce)/HZSM-5), 1.332% of ether was formed, which was the least amount of ether produced in this study. The percentages of ester, aldehyde, and ketone were 9.858%, 3.261%, and 1.883%, respectively. Using this catalyst, 42.748% of alcohol was formed.

The third Lanthanum-Cerium-modified zeolite catalyst ((7.5%La-2.5%Ce)/HZSM-5) produced the most ether and hydrocarbons out of all of the catalysts made for this study (HZSM-5, (2.5%La-7.5%Ce)/HZSM-5, and (5%La-5%Ce)/HZSM-5). The percentages were 5.831% and 42.963%, respectively. The hydrocarbons (alkane and alkene) were distributed so that each contained 5.33% alkane and 37.63% alkene. On the other hand, we found that this catalyst produced the least amount of ester and ketone out of all of the catalysts that were synthesized; the percentages were 3.03% and 1.623%, respectively. The amount of aldehyde formed was 4.546% and the amount of alcohol made was 35.259%.

Furthermore, the parent HZSM-5 catalyst formed the highest amounts of the oxygenated compounds (esters and aldehydes) and the lowest amounts of the hydrocarbons (alkanes and alkenes). In this work, (7.5%La-2.5%Ce)/HZSM-5 produced the highest amounts hydrocarbons. The results also reveal that the catalyst (7.5%La-2.5%Ce)/HZSM-5 produced the least amount of ester and ketone. In this study, the highest amounts of alcohol were produced for (2.5%La-7.5%Ce)/HZSM-5.

As a result, the catalytic deoxygenation of algal HO for the parent HZSM-5 zeolite and Lanthanum-Cerium-modified zeolite catalysts ((2.5%La-7.5%Ce)/HZSM-5), ((5%La-5%Ce)/HZSM-5), and ((7.5%La-2.5%Ce)/HZSM-5) can produce oleochemicals, especially hydrocarbons and alcohol groups. Those are important compounds that can be used to produce biofuels.

Based on Figure 12b, there is no clear link between the physical and chemical properties of the yield of the hydrocarbon percentages. The micropore areas and surface areas of the parent HZSM-5 catalyst were the highest of all of the Lanthanum-Cerium-modified zeolites in this study. However, the yield percentages of the hydrocarbons from this catalyst (HZSM-5) were the lowest (21.833%). Meanwhile, the catalyst with the fewest micropores, (2.5%La-7.5%Ce)/HZSM-5, did not produce the highest hydrocarbon yield percentage (24.50%). As mentioned before, the micropore areas and surface areas of all of the Lanthanum-Cerium-modified zeolites were smaller than those of the HZSM-5 zeolite (see Table 2). For (7.5%La-2.5%Ce)/HZSM-5, a yield of 42.963% of hydrocarbons was found to be the highest.

It can be determined that the physical properties of the Lanthanum-Cerium loading percentages on the parent HZSM-5 are not the primary cause for the generation of hydrocarbons. This conclusion is consistent with the explanation by Zaki et al. that these physical properties, such as the surface area and micropore area, have no effect on the effectiveness of the parent catalyst (HZSM-5) for the production of olefins after it has been modified with metals, such as Cu/HZSM-5 and Ni/HZSM-5 [40].

Figure 12c shows that the acidity of all of the synthesized catalysts in this study is related to the percentage of hydrocarbon yield. It can be seen that the parent HZSM-5, which had the most acid sites (0.740 mmol/g), had the lowest hydrocarbon yield, while the (2.5%La-7.5%Ce)/HZSM-5 catalyst, which had the fewest acid sites (0.524 mmol/g) in this study, did not have the highest hydrocarbon yield. As mentioned previously, the total acid cites for all the Lanthanum-Cerium-modified zeolite catalysts were fewer than for the original HZSM-5 (refer to Figure 8 and Table 3). The total acid increased as the Lanthanum loading percentage on the parent HZSM-5 increased and the Cerium loading percentage decreased for all of the composite catalysts in this study ((2.5%La-7.5%Ce)/HZSM-5, (5%La-5%Ce)/HZSM-5, and (7.5%La-2.5%Ce)/HZSM-5), which were 0.524, 0.584, and 0.602 (mmol/g), respectively.

However, the performance of catalytic deoxygenation in producing hydrocarbons changed depending on how much Lanthanum and Cerium were added to the parent HZSM-5.

In general, all of the composite Lanthanum-Cerium-modified zeolite catalysts produced higher amounts of hydrocarbons compared to the parent HZSM-5 (see Figure 12c). On the other hand, for all of the composite catalysts in this study, increasing the Lanthanum loading percentage and decreasing the Cerium loading percentage on the parent HZSM-5 increases the total acidity and the yield percentages of hydrocarbons, as shown in Figure 12c. As a result, as compared to the HZSM-5 parent catalyst, the production of hydrocarbons increased as the total acidity decreased.

#### 3.2.3. Overall Reaction Pathways Proposed

Figure 13 shows the proposed overall reaction pathways for the catalytic deoxygenation of algal HO for all of the synthesized catalysts in this study (HZSM-5, (2.5%La-7.5%Ce)/HZSM-5, (5%La-5%Ce)/HZSM-5, and (7.5%La-2.5%Ce)/HZSM-5). As mentioned before, all of these experiments were completed in the batch reactor at the same operating conditions: 300 °C, 6 h, 7 bar of N_2_, 1000 rpm, and 15% (wt.%).

Several different studies [17,24,108,109,110,111] were used to come up with the suggested reaction pathways. In addition to the results of this study for several catalysts (see Table 5), the catalytic deoxygenation of algal HO made products, such as non-oxygenated compounds (alkane and alkene) and oxygenated compounds (aldehyde, ketone, ester, ether, and alcohol).

In this study, theories were made about how these products were made, and Figure 13 shows those theories. The type of catalyst used had a great effect on how the reaction went. As a result, it was important to figure out how each product was made during the catalytic deoxygenation of the algal HO so that the effects of the catalysts on the yield of these products could be evaluated in a realistic way.

According to the literature [108,109,110], the following chemical reactions (1–4) are common: Decarbonylation
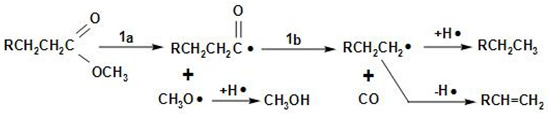
Demethylation (2a, 2b) and decarboxylation or deketonization (2c, 2d) upon the ester bond cracking
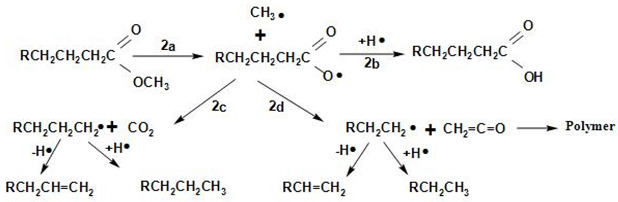
Cracking of the hydrocarbons formed in the reactions 1b, 2c, and 2d
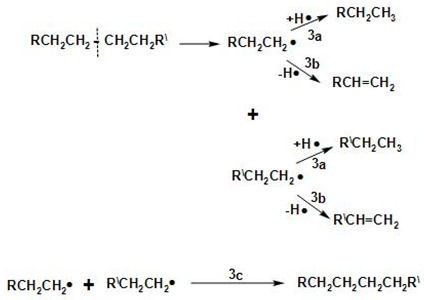
Cracking of fatty acid methyl esters (FAMEs)
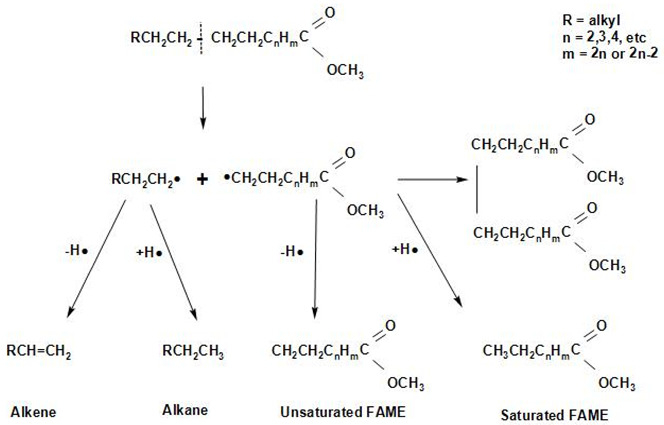


At first, FAMEs go through a thermal deoxygenation transformation mechanism described by Seames et al. [109], where FAMEs are changed into intermediates of an aldoketone (R1), acrolein (R2), and fatty acid (R3) [24], however, these intermediates are so unstable that they were not found in our results in this study (see Table 5).

The aldoketone is unstable and quickly changes into hydrocarbon radicals (R^•^ or R^\^^•^) or aldehyde through decarbonylation (R4 and R5) or hydrogenation (R6). By hydrogenation, the aldehyde made by this reaction (R6) could also be turned into alcohol (R7).

Acrolein breaks down into olefins, such as propylene (R8) and coke, when it polymerizes (R9).

The fatty acids formed by R3 are deoxygenated by either decarboxylation (R10) to produce an alkane or decarbonylation (R11) to form alcohol. The alcohol formed by R11 and R7 could then be turned into an alkene by dehydration (R12, R13).

The products of ethers that were shown in this study (see Table 5), such as Disparlure and tridecyl-Oxirane, can be mostly attributed to the bimolecular dehydration of alcohols caused by R7 and R11 to produce ethers R14 and R15, as shown in Table 5. 

Table 5 shows the ketonic compounds, which are a type of ketone that can be produced by the ketonization process. As shown in R16, two fatty acids were combined in a dehydration process to make the anhydride. Then, the anhydride was turned into a ketone in a ketonization process while CO_2_ was removed (R17). This study did not detect any symmetric ketones. This means that symmetric ketones could be broken down into olefins (R18) and non-identical ketones through γ-hydrogen transfer (R19).

The paraffins and olefins that were produced were then broken up into smaller pieces (R20–R21).

In addition to ketones, we observed the formation of aldehydes, such as Tetradecanal and Octadecanal. However, there have not been many reports of aldehyde being produced in thermal or catalytic cracking reactions. Aldehyde has only been found to form in thermal cracking at temperatures above 600 °C [112,113], which is much higher than the reaction temperature in this study (300 °C). These differences show that the aldehyde formation seen in this study is unique to the reactions on zeolite catalysts with enhanced hydrogen transfer activity and is different from thermal cracking. Even in a hydrogen-free atmosphere, algal HO deoxygenation on zeolite catalysts with increased hydrogen transfer activity still produces aldehydes as intermediates and involves hydrodeoxygenation. These signs show that there were reduction, hydrogenation, and dehydration reactions.

Reductions, such as R14, and hydrogenation, such as R6, can be explained by a mechanism called “hydrogen transfer” in which active hydrogen species are released when intermediate hydrocarbons are broken down and transferred to fatty acids and aldehydes. The reaction paths to produce alkanes through hydrogenation are the same as those proposed for the hydrocracking reactions of triglycerides on metal oxide catalysts [114]. This shows that hydrogen transfer reactions during catalytic deoxygenation have strong hydrogenation activity and that algal HO can be turned into hydrocarbons in an efficient way without using a hydrogen atmosphere.

When the hydrocarbon yield was looked at, it was found that C_11_, C_12_, C_14_, and C_19_ hydrocarbons were more selective in almost all processes. This could be because of the fatty acids in the oil esters. They have a lot of oleic (C18:1) and palmitic (16:0) acids, which can cause this behavior. C_11_–C_14_ hydrocarbons can be produced from oleic acid by β-scission to the double bond, followed by fatty acid deoxygenation [111] and radical reactions (disproportionation, condensation, and β-scission to the radical carbon). In addition, C6, C_8_–C_13_ can be made by deoxygenating oxygenating fragments made by β-scission to a double bond. This can be achieved through similar radical reactions. Moreover, cyclic and aromatic hydrocarbons can be made from olefinic fragments, which are also formed by acid oleic β-scission [111].

#### 3.2.4. The Distribution of Carbon Numbers

Based on the results shown in Figure 12a and Table 5, when looking at the algal HO conversion into product chemical composition groups, it can be seen that the catalytic deoxygenation for the synthesized catalysts conditions of an initial 7 bar N_2_ pressure, 300 °C, 6 h, 15% catalyst-to-HO weight ratio, and 1000 rpm showed the high conversion of HO and the interesting chemical groups. In this study, the results are expressed in terms of carbon number and product content so that the carbon number distributions of the liquid products from these operating conditions could be compared to those of all of the synthesized catalysts.

As previously stated, there were seven groups of liquid products: two for compounds without oxygen (alkanes and alkenes) and five for compounds with oxygen (esters, ethers, aldehydes, ketones, and alcohols). The main results that were found were non-oxygenated compounds (alkenes) and oxygenated compounds (alcohols). Small amounts of alkanes, esters, ethers, aldehydes, and ketones were also found. Figure 12a and Table 5 show the compounds and contents of the algal HO. The main components of the algal HO are made up of three chemical groups: alkane (11.538%), esters (61.123%), and alcohol (phytol) (21.62%). Based on the GC-MS results (Figure 12a and Table 5), the yield of phytol increased in all of the experiments performed in this study, while the yield of alkanes and esters decreased and were turned into other compounds (oxygenated compounds and non-oxygenated compounds). In conclusion, the phytol was not a reactant compound but it can be thought of as a product compound in this study. The alkanes and the fatty acid methyl esters can be thought of as reactant compounds.

As shown in Figure 14a, the esters made up most of the algal HO (61.123%). Most of them had C_19_ carbon atoms (42.776%), where those distributed 15.084% in 9,12,15-Octadecatrienoic acid, methyl ester, (Z,Z,Z)-(C_19_H_32_O_2_) and 27.692% in 9,12-Octadecadienoic acid, methyl ester, (Z,Z,Z)-(C_19_H_34_O_2_). The other esters were made up of 15.325% Hexadecanoic acid, methyl ester (C_17_), 1.728% 6-Octen-1-ol, 3,7-dimethyl-, formate (C_11_), and 1.294% Di-n-octyl phthalate (C_24_). In Hexacosane (C_26_H_54_), the alkane group was shown in C_26_ carbon atom (26) at a yield of 11.538%. In phytol (C_20_H_40_O), the alcohol chemical group was made up of C_20_ carbon atoms at a yield of 21.620%.

For the liquid product of the catalytic deoxygenation of algal HO for the parent HZSM-5 zeolite (Figure 14b), most of the products were in the alcohol group (49.549%). Most of the alcohols had C_20_ carbon atoms (20) in phytol (C_20_H_40_O) at percentages of 40.859% and 8.69% in C_15_ carbon atoms (15) in 1-Dodecanol, 3,7,11-trimethyl-(C_15_H_32_O). The other main components were non-oxygenated compounds (hydrocarbons), which were found in alkanes and alkenes. The total amount of hydrocarbons was 21.833% and most of them were found in 5-Ethyl-1-nonene (C_11_H_22_) at a percentage of 17.048% and in Tetradecane (C_14_H_30_) at a percentage of 4.785%. The other parts were made up of small amounts of esters, ethers, aldehydes, and ketones. The total amount of esters was 12.64 percent, which was made up of 5.41% of C_17_ esters, 1.88% of C_16_ esters, and 5.34% of C_19_ esters. These were Hexadecanoic acid, methyl ester (C_17_H_34_O_2_), Carbonic acid, and butyl undec-10-enyl ester (C_16_H_30_O_3_). The ethers group was found in 4.93% of those, mostly 2.847% (C19), 1.468% in Disparlure (C_19_H_38_O), 1.379% in Tetrahydropyran 12-tetradecyn-1-ol ether (C_19_H_34_O_2_), and 2.083% (C_15_) in Oxirane, tridecyl-(C_15_H_30_O). Tetradecanal had 6.471% of the aldehyde group C_14_ in it (C_14_H_28_O). At 2.394% in C_18_ of 2-Pentadecanone, 6,10,14-trimethyl, the ketones group was found in the product C_18_H_36_O.

For the liquid product of the catalytic deoxygenation of algal HO for the (2.5%La-7.5%Ce)/HZSM-5 zeolite (Figure 14c), the products mostly consisted of non-oxygenated compounds (hydrocarbons) at a total percentage of 24.50%, which was mostly distributed in the alkene group with a C_11_ carbon atom of 11.592% that was 5-Ethyl-1-nonene (C_11_H_22_) and C_12_ in 3.339% of 1-Undecene, 8-methyl-(C_12_H_24_). Supraene (C_30_H_50_) contains C_30_ in 3.371% of it and the alkane group with Nonadecane (C_19_H_40_) contains C_12_ in 3.113% and Bicyclo [3.1.1]heptane, 2,6,6-trimethyl-,(1.alpha.,2.beta.,5.alpha.) with C_10_ in 3.085%. The other main product of this catalyst ((2.5%La-7.5%Ce)/HZSM-5) is the alcohol group, which has a percentage of 50.5% and is mostly made up of C_20_ carbon atoms of 22.38% Phytol (C_20_H_40_O), C_19_ carbon atoms of 12.299% 2-Methyl-Z,Z-3,13-octadecadienol (C_19_H_36_O), C_22_ in 1.415% of 1,22-Docosanediol (C_22_H_46_O_2_), and C_14_ in 14.406% of Cyclododecanol, 1-ethenyl-(C_14_H_26_O). The esters group was found. A total of 8.152% of the product included C_19_ in 2.006% of 6-Octadecenoic acid, methyl ester, (Z)-(C_19_H_36_O_2_) and 1.063% of Cyclopentanetridecanoic acid, methyl ester (C_19_H_36_O_2_), and C_17_ in 5.083% of Hexadecanoic acid, 2-hydroxy-, methyl ester (C_17_H_34_O_3_). The ether group in 2-Furanmethanamine, tetrahydro-(C_5_) was 4.613% (C_5_H_11_NO). Aldehyde was present in the liquid product at a concentration of 1.52% (C_14_) of Tetradecanal (C_14_H_28_O). The ketone group accounted for 5.861% in C_26_ of B(9a)-Homo-19-norpregna-9(11),9a-dien-20-one, 3-(dimethylamino)-4,4,14-trimethyl-, (3.beta.,5.alpha.)-(3.beta.,5.alpha.)-(C_26_H_41_NO).

For the liquid product of the catalytic deoxygenation of algal HO for the (5%La-5%Ce)/HZSM-5 zeolite (Figure 14d), the products primarily consisted of non-oxygenated compounds (hydrocarbons) at a total percentage of 35.149% and was predominantly distributed in the alkene group with C_11_ carbon atom of 24.827% that was 5-Ethyl-1-nonene (C_11_H_22_) and 6.010% in C_12_ of 1-Undecene, 8-methyl-(C_12_H_24_), while the alkane group with 4.312% in C_19_ was Nonadecane (C_19_H_40_) (C19H40). The second principal product of this catalyst ((5%La-5%Ce)/HZSM-5) is the alcohol group at a percentage of 42.748%, mostly distributed in the C_20_ carbon atom of 34.091% Phytol, 7.341% 3,7,11,15-Tetramethyl-2-hexadecen-1-ol (C_20_H_40_O), and 1.316% 1,1’-Bicyclopenta-1,1-diol (C_10_) (C_10_H_18_O_2_). The distribution of the ester group in the product was 3.184% (C_15_), 3.847% (C_19_), 1.011% (C_24_), and 1.816% (C_16_). These consisted of Tridecanoic acid, 12-methyl-, methyl ester (C_15_H_30_O_2_); 11-octadecenoic acid, methyl ester (C_19_H_36_O_2_); fumaric acid, 2,4-dimethylpent-3-yl tridecyl ester (C_24_H_44_O_4_); and 2,2-Dimethylpropanoic acid, 2,6-dimethylnon-1-en-3-yn-5-yl ester (C_16_H_26_O_2_), respectively. The ether group in 1-(ethenyloxy)-octadecane (C_20_) was 1.332% (C_20_H_40_O). The liquid product included an aldehyde group at a concentration of 3.261% in (C_14_) of Tetradecanal (C_14_H_28_O). The ketone group included 0.998% of 2(4H)-Benzofuranone, 5,6,7,7a-tetrahydro-4,4,7a-trimethyl-, (R)-(C_11_H_16_O_2_) in (C_11_) and 0.885% of Benz[e]azulene-3,8-dione, 5-[(acetyloxy)methyl] in C_19_ (-3a,4,6a,7,9,10,10a, 10b-octahydro-3a, 10a-dihydroxy-2, 10-dimethyl-, (3a.alpha., 6a.alpha., 10.beta., 10a.beta., 10b.beta.)-(+)-(C_19_H_24_O_6_)).

For the liquid product of the catalytic deoxygenation of algal HO for the (7.5%La-2.5%Ce)/HZSM-5 zeolite (Figure 14e), the products primarily consisted of the alcohol group (35.259%), primarily distributed in C_20_ carbon atoms (20) in phytol (C_20_H_40_O) in 26.352%, at 5.698% in 3,7,11,15-Tetramethyl-2-hexadecen-1-o (C_16_H_34_O), in C_12_ carbon atoms (12) at a percentage of 1.947% in trans-2-Dodecen-1-ol (C_12_H_24_O), and at 1.262% in C_16_ of 1-Decanol, 2-hexyl-(C_16_H_34_O). The remaining major components were non-oxygenated compounds (hydrocarbons) at a total percentage of 42.963%, which were primarily distributed as alkene C_11_ at 30.055%, 7.578% in C_12_ and alkane C_19_, namely 5-Ethyl-1-nonene (C_11_H_22_), 1-Undecene, 8-methyl-(C_12_H_24_), and Nonadecane (C_19_H_40_), respectively. Hexadecanoic acid, methyl ester (C_17_H_34_O_2_) and Butyl 9-tetradecenoate (C_18_H_34_O_2_) constituted 1.965% (C_17_) and 1.065% (C_18_) of the total quantity of esters, respectively, which amounted to 3.03%. The ethers group was detected in 5.831% of those mostly dispersed in 1.504% (C_11_) of 2H-Pyran, 2-[(2-furanylmethoxy)methyl]tetrahydro-(C_11_H_16_O_3_) and 4.327% (C_39_) of 9-Octadecene, 1-[3-(octyloxy)propoxy]-, (Z-) (C_39_H_78_O_2_). The overall percentage of the aldehyde group in 2-Heptadecenal (C_17_) was 4.546% (C_17_H_32_O). A total of 1.623% of C_14_ of 2(1H)-Naphthalenone, octahydro-4a, 5-dimethyl-3-(1-methylethyl)-, (3. alpha.,4a.beta.,5.beta.,8.alpha.) included a ketone group (C_14_H_24_O).

#### 3.2.5. Outstanding Bio-Based Chemical Products

In Equation (1), the percentage yield of the selected outstanding compounds of hydrocarbons (alkanes and alkenes) and alcohol groups from the catalytic deoxygenation of algal HO for the parent HZSM-5 catalyst and Lanthanum-Cerium-modified zeolite catalysts was calculated and is shown in Table 6.

Table 6 shows that non-oxygenated compounds (hydrocarbons) and oxygenated compounds (alcohols) had higher yield percentages (˂20%) during catalytic deoxygenation reactions of algal HO using the parent HZSM-5 and Lanthanum-Cerium-modified zeolites as catalysts. Consequently, these catalysts are interesting for catalyzing the deoxygenation of FAMEs even at a low Nitrogen pressure (7 bar), which means they are cheaper.

Table 6 and Figure 15 show that the yields of hydrocarbons from the catalytic deoxygenation of algal HO from Lanthanum-Cerium catalysts were higher than the yields of hydrocarbons from the parent HZSM-5 catalyst. In more detail, the total yield of hydrocarbons from the catalytic deoxygenation of algal HO for the parent HZSM-5, (2.5%La-7.5%Ce)/HZSM-5, (5%La-5%Ce)/HZSM-5, and (7.5%La-2.5%Ce)/HZSM-5 were 21.833%, 24.50%, 35.149%, and 42.963%, respectively. The highest yield of hydrocarbons was 42.963% and the highest conversion of algal HO was 100% (see Table 6 and Figure 16). This was completed in a batch reactor at 300 °C for 6 h with 7 bar of initial inert N_2_ gas and a catalyst-to-HO ratio of 15% (by weight).

In this study, the products from the parent HZSM-5 had the lowest yield of hydrocarbons (21.833%) and the lowest conversion (94.589%) of the algal HO. This could be because of the number of other compounds with oxygen. For example, the fact that the catalytic deoxygenation of algal HO makes fewer hydrocarbons than the parent HZSM-5 catalyst might be because it makes more alcohols, esters, aldehydes, ketones, and ethers (refer to Table 5).

In this research, the yield percentages of alkanes are much lower than those of alkenes. This may be because the catalytic deoxygenation of algal HO only produced a small amount of hydrogen, which was not enough to fill the double bonds during these reactions. However, this study was completed under an initial pressure of 7 bar, which was the pressure of the inert gas nitrogen.

In conclusion, as was explained in the Ammonia TPD discussion, adding Lanthanum-Cerium to the parent HZSM-5 zeolite has a great effect on the total acid sites, especially the acidic Bronsted sites. However, this area of strong acid sites (the Bronsted sites) is thought to be the main catalytic center and include the dominant acid sites during catalytic deoxygenation reactions [32], whereas the number of weak acid sites (the Lewis sites) was deemed to be not catalytically important but it is assumed to effectively influence the proton mobility in zeolites as shown in the investigation of Zaki et al. [40]. 

Table 7 shows the results of different studies on the catalytic deoxygenation process using different types of catalysts to make hydrocarbons from different feeds (fatty acids or FAMEs). The operating conditions of those studies (reactor type, time, temperature, and the initial value of the charged gas pressure) are very close to the operating conditions used in this study. However, our results show that Lanthanum-Cerium-modified zeolite showed deoxygenating activity under a low pressure of an inert gas (Nitrogen). This could mean that the cost of making hydrocarbons without using hydrogen gas would be much lower.

Many researchers have discussed catalytic cracking deoxygenation using different catalysts and reactants under an initial H_2_ pressure or N_2_ inert gas pressure. Studies on catalytic deoxygenation with the initial pressure of an inert gas are a lot fewer than those on the same process with the initial pressure of hydrogen gas. In the meantime, there is no other research that is similar to this study’s catalytic deoxygenation of algal HO using these catalysts. As a result, this study’s operating conditions (such as the type of reactor, temperature, time, fatty acids or FAMEs reactants, and the initial pressure of the pumped gas) were compared to similar catalytic studies (refer to Table 7).

Paying the most attention to catalytic deoxygenation under the pressure of H_2_ gas, Sousa et al. (see Table 7) talked about the catalytic deoxygenation of palm kernel oil and the hydrolysis of palm kernel oil for HBeta zeolite under 10 bar of H_2_ as the initial pressure. The yields of hydrocarbons were 82 ± 3% and 24 ± 9%, respectively. However, they explained that the yields of hydrocarbons for the catalytic deoxygenation of olein oil and hydrolyzed olein oil were 43 ± 3% and 98 ± 4%, respectively, using the same catalyst and the same operating conditions as shown in Table 7. They demonstrated that the type of reactants (the size of the reactant molecule and the length of the carbon chain of the reactants) has a significant impact on the number of hydrocarbons produced using the same catalyst and operating conditions [13].

Meller et al. looked at how the type of solvent and the temperature affected the catalytic deoxygenation of hydrolyzed castor oil to make hydrocarbons using a Pd/C catalyst and 25 bar of H_2_ at the start (refer to Table 7). According to Table 7, the type of solvent used and the temperature of the reaction have a significant impact on the number of hydrocarbons produced [14].

Peng et al. looked at the catalytic deoxygenation of stearic acid for 10%Ni/HZSM-5 in the presence of a solvent (dodecane) under an initial pressure of 40 bar of H_2_. At 260 °C for 8 h, the total yield of hydrocarbons was about 56% (refer to Table 7). Peng et al. also conducted a study on the catalytic conversion of microalgae oil for 10%Ni/ZrO_2_ at 270 °C with 40 bar of initial H_2_ pressure, and no solvent. They searched for the effect of reaction time on the total yield of hydrocarbons under the same operating conditions. The total yield of hydrocarbons after 6 h was 72% and after 4 h it was 61% [16] (as presented in Table 7).

Duongbia et al. wrote about the effect of solvent on the catalytic deoxygenation of palmitic acid for Ni/LY char catalyst under 30 bar of initial H_2_ pressure at 300 °C and 5 h. The highest yield of hydrocarbons was 12.75 % when hexane was used as a solvent [17] (refer to Table 7).

Considering catalytic deoxygenation under pressure from an inert gas, Snare et al. [18] discussed the catalytic deoxygenation of methyl oleate in an initial reaction atmosphere of H_2_ and Ar. They showed that the type of initial gas pressure has a great effect on the conversion percentage and the number of hydrocarbons made under the same catalyst, temperature, feed-to-catalyst ratio, reaction time, the highest conversion percentage, and the selectivity of the hydrocarbons [18].

Morgan et al. studied the catalytic deoxygenation of soybean oil with 20% Ni/Al_2_O_3_ under 7 bar of inert gas (N_2_) at 350 °C. The highest hydrocarbon yield percentage was 79.5% and the conversion was 74% [26] (refer to Table 7).

Hollak et al. showed that the total selectivity of hydrocarbons was 35% in the catalytic deoxygenation of stearic acid for Pd/Al_2_O_3_ under 7 bar of nitrogen inert gas pressure at 350 °C for 6 h with a 43% conversion percentage [27] (refer to Table 7).

In general, the Lanthanum-Cerium-modified zeolites produced more non-oxygenated compounds than the algal HO for the parent HZSM-5 when it came to catalytic deoxygenation. This finding agrees with what Li, J., et al. found when they used HZSM-5 and 5%Fe/HZSM-5 to catalyze the liquefaction of cellulose in the presence of the solvent (n-heptane) at 350 °C in a batch reactor. They found that the parent HZSM-5 produced higher yields of oxygenated compounds and lower yields of non-oxygenated compounds [28].

In conclusion, the above studies show that the percentage of hydrocarbons produced by catalytic deoxygenation depends on several factors, such as temperature, use of solvent, type of solvent, type of reactant, feed-to-catalyst ratio, reaction time, and the type of initial pressure that is pumped into the reactor before the reaction happens.

Paying the most attention to the alcohol compounds made in this study, Figure 15 shows that the alcohol yields from the catalytic deoxygenation of algal HO for all of the Lanthanum-Cerium-modified zeolite catalysts were less than the alcohol yields from the parent HZSM-5 catalyst, except for (2.5%La-7.5%Ce)/HZSM-5. In more detail, the total yield of alcohol from the catalytic deoxygenation of algal HO for the parent HZSM-5, (2.5%La-7.5%Ce)/HZSM-5, (5%La-5%Ce)/HZSM-5, and (7.5%La-2.5%Ce)/HZSM-5 was 49.549%, 50.5%, 42.748%, and 35.259%, respectively. The most alcohols were made when (2.5%La-7.5%Ce)/HZSM-5 was used. The highest yield value was 50.5%. In this study, the catalytic deoxygenation of algal HO in the batch reactor at 300 °C for 6 h with 7 bar of initial inert N_2_ gas and a catalyst-to-algal HO ratio of 15% (wt.%) led to the lowest conversion rate of 94.589% for the parent HZSM-5 (see Table 6 and Figure 15).

In conclusion, the parent HZSM-5 catalyst had the lowest hydrocarbon yield percentage. Additionally, (7.5%La-2.5%Ce)/HZSM-5 had the lowest percentage of alcohol yield and the highest percentage of hydrocarbon yield. This could be because HZSM-5 (0.214 mmol/g) has more strong acid sites than (7.5%La-2.5%Ce)/HZSM-5 (0.154 mmol/g). Moreover, strong acid sites play a key role in the catalytic deoxygenation of oxygenated compounds [32]. Therefore, a small number of strong acid sites could make it easier for oxygenates to be turned into hydrocarbons.

Placing the most effort into studies about the catalytic deoxygenation process, which is about making oxygenated compounds, such as alcohol, most studies on catalytic deoxygenation have discussed the yields of non-oxygenated compounds (hydrocarbons). There are not as many studies on the production of oxygenated compounds during catalytic deoxygenation as there are on the production of non-oxygenated compounds (hydrocarbons). In the meantime, as mentioned previously, there is no other study in the literature that is similar to the catalytic deoxygenation of algal HO conducted in this study using these catalysts. Therefore, studies of close catalytic cracking were compared in terms of the reactants and the percentages of alcohol that were made during these reactions.

Duongobia et al. wrote about how the reaction time affects the yield percentage of alcohol when palmitic acid is treated with a Limonite catalyst. At 5 h and 3 h, the yield percentages of alcohol were 51.84% and 38.35%, respectively [17].

J. Li et al. looked at the catalyzed liquefaction of cellulose with the help of glycerol for HZSM-5 and 5% Fe/HZSM-5 under the same operating conditions. They found that the modification of the parent HZSM-5 with Fe metal gave a lower yield percentage of alcohol (20%) than the parent HZSM-5 (26%) [28].

Rozmyslowics et al. looked at the catalytic deoxygenation of Lauric acid for 5% Pd/C under the same operating conditions (except for the initial pressure gas). They found that the type of initial pressure had a great effect on the amount of alcohol that was made. The total amount of alcohol made under 20 bar of H_2_ was 9% but none was made under 20 bar of Ar [19,20]. Furthermore, Rozmyslowics et al. showed the effect of the reaction time in the catalytic deoxygenation of stearic acid for 4% Ru/TiO_2_ under the same operating conditions (except for the reaction time). The yield percentage of alcohol was 20% at 1 h but none was made at 6 h [29]. On the other hand, they showed that the type of catalyst has a great effect on the amount of alcohol that is made during the catalytic deoxygenation of stearic acid. Two experiments were performed with the same operating conditions (except for the type of catalyst. The total amount of alcohol made from 4% Re/TiO_2_ was 81% but there was no alcohol made from 4% Ru/TiO_2_ [29].

In the catalytic cracking of soybean oil for a composite catalyst (35% ƔAl_2_O_3_/CaO) in a fixed-bed reactor, Zheng et al. [115] found that the highest yield of alcohol was 12.3%.

Balasundram et al. claimed that the yield percentage of alcohol in the catalytic pyrolysis of sugarcane bagasse for the parent HZSM-5 and 1% Ce/HZSM-5 was about 14% and 5%, respectively [39].

In conclusion, the above studies show that the percentage of alcohols produced by different reactions, such as catalytic deoxygenation, depends on several factors, such as temperature, type of catalyst, reaction time, and the type of initial pressure that is pumped into the reactor before the reaction happens.

#### 3.2.6. Liquid Product Characterization

The liquid products were obtained from the catalytic deoxygenation of the algal HO for the parent HZSM-5 and the Lanthanum-Cerium-modified zeolite catalysts at the same operating conditions: a 300 °C reaction temperature for 6 h with a 15% catalyst-to-HO weight ratio at 1000 rpm and with 7 bar N_2_ in the batch reactor. Table 8 shows the elemental compositions of liquid products, which were calculated using Equation (3).

Compared to the algal HO, the carbon and hydrogen weight percentages of the products of the catalytic deoxygenation for (2.5%La-7.5%Ce)/HZSM-5, (5%La-5%Ce)/HZSM-5, and (7.5%La-2.5%Ce)/HZSM-5 increased, while the oxygen weight percentages decreased. The results of the parent HZSM-5 show that the weight percentage of carbon increased, while the weight percentages of hydrogen and oxygen decreased. The Jafarian study [116] found that the higher heating value (HHV) increased when the amount of carbon and hydrogen increased and the amount of oxygen decreased. As shown in Table 8, compared to the HHV (MJ/kg) of HO (32.377), the HHV of the catalytic deoxygenation liquid products for the parent HZSM-5, (2.5%La-7.5%Ce)/HZSM-5, (5%La-5%Ce)/HZSM-5, and (7.5%La-2.5%Ce)/HZSM-5 were higher by 33.230, 33.738, 34.036 MJ/kg, respectively. As calculated from the O/C molar ratios using Equation (5), the degree of deoxygenation percentage (DOD%) of the liquid products from the modified Lanthanum-Cerium zeolite (7.5%La-2.5%Ce)/HZSM-5 was higher than the DODs of the liquid products from other synthetic catalysts in this work. If oxygen is taken out of the fuel, the viscosity and acidity of the fuel may be improved [117]. However, compared to fossil crude oil, the HHVs of all of the products of the catalysts were low [118].

The atomic ratios of H/C and O/C are shown in Table 8, which is a van Krevelen diagram (Figure 16). Compared to algal HO, the ratio of H/C in the liquid products of all Lanthanum-Cerium-modified zeolite catalysts increased, while the ratio of O/C decreased. The H/C and O/C atomic ratios of the parent HZSM-5 zeolite were lower than those of the raw algal HO. The (7.5%La-2.5%Ce)/HZSM-5 had the highest H/C ratio of 1.994 and the lowest O/C ratio of 0.031 among the Lanthanum-Cerium-modified zeolites. Even though the H/C ratios of the liquid products were high, the O/C ratios stayed high when compared to the O/C ratio of fossil crude oil (~0). It should be noted that the highest value of DOD% was 62.109% for (7.5%La-2.5%Ce)/HZSM-5.

Comparing the results of this study with those of a previous one [17], which looked at the hydrotreatment of palmitic acid for a Ni/LY catalyst under 30 bar H_2_ at 300 °C in a batch reactor, the highest DOD% was 65.15%, which is low compared to the amount of hydrogen used. The highest H/C and HHV were also 2.03 and 32.4 MJ/Kg, respectively, while the highest H/C and HHV in this study were 1.994 and 34.362 MJ/Kg, respectively.

## 4. Conclusions

Using a batch reactor, the performance of composite Lanthanum-Cerium -impregnated HZSM-5 for the catalytic conversion of algal HO to non-oxygenated and oxygenated compounds was evaluated. The physical alterations observable in the textural qualities were caused by the loading percentages of lanthanum and cerium. From the physical qualities, no direct link can be inferred toward increasing the hydrocarbon yield. The addition of Lanthanum-Cerium into HZSM-5 zeolite at varying percentages improved the acid sites required for algal HO conversion. The conversion percentages of algal HO using all Lanthanum-Cerium-modified HZSM-5 (in the range of 98.035% to 100%) were greater than the conversion percentage utilizing the original HZSM-5 percentage (94.589%). The incorporation of Lanthanum-Cerium into HZSM-5 has the added impact of producing a greater hydrocarbon yield (24.5–42.963%). In general, the rising performance of catalysts in converting algal HO to hydrocarbons is as follows: (7.5%La-2.5%Ce)/HZSM-5 > (5%La-5%Ce)/HZSM-5 > (2.5%La-7.5%Ce)/HZSM-5 > HZSM-5. The alcohol products from the Lanthanum-Cerium-modified zeolite ranged from 35.259% to 50.5%, while the alcohol products from the HZSM-5 parent were 49.549%. The DOD percentages for liquid products from Lanthanum-Cerium-modified HZSM-5 catalysts ranged from 46.755 to 62.109%, while the DOD% for liquid products from the pure HZSM-5 catalyst was 44.235%. The HHV of the liquid products from all of the Lanthanum-Cerium-modified HZSM-5 catalysts (ranging from 33.738 to 34.362 MJ/Kg) was greater than the HHV of the liquid product from the parent HZSM-5 (33.230 MJ/Kg). The catalyst (7.5%La-2.5%Ce)/HZSM-5 provided the greatest yield of hydrocarbons, highest HHV, and highest DOD%, which were 42.963%, 34.362 MJ/Kg, and 62.109%, respectively, among all the synthesized catalysts in this investigation.

## Figures and Tables

**Figure 1 molecules-27-08018-f001:**
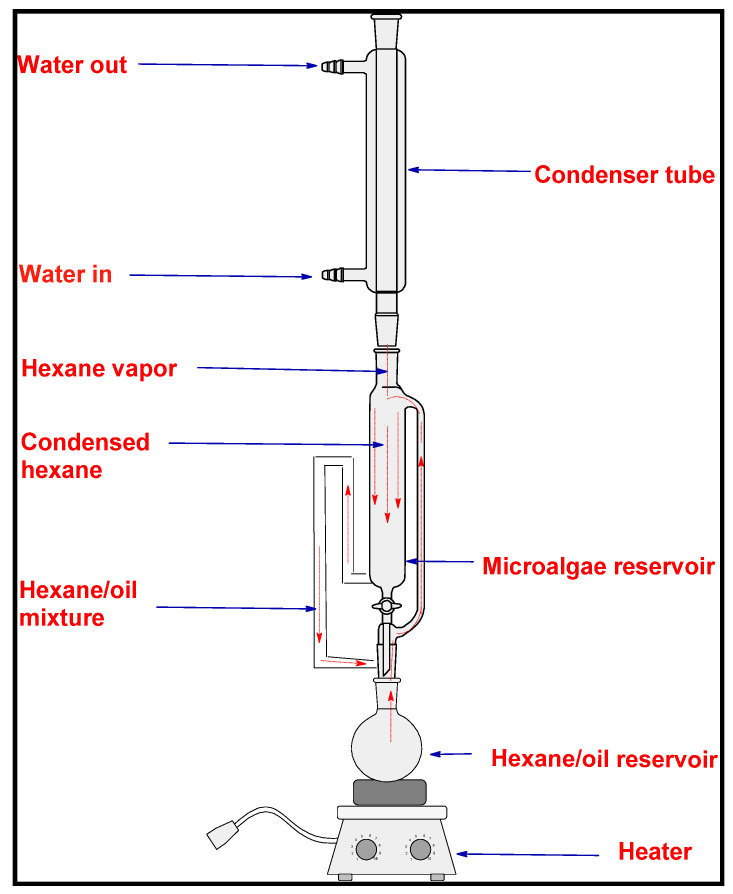
Scheme of the soxhlet extractor used to extract the crude oil from *Chlorella vulgaris* microalgae powder.

**Figure 2 molecules-27-08018-f002:**
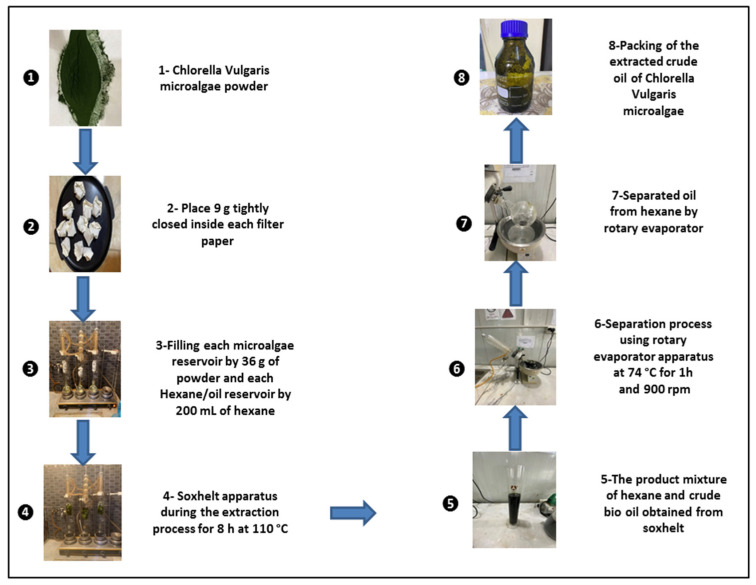
Experimental images for the extraction procedure for the crude oil from *Chlorella vulgaris* microalgae powder.

**Figure 3 molecules-27-08018-f003:**
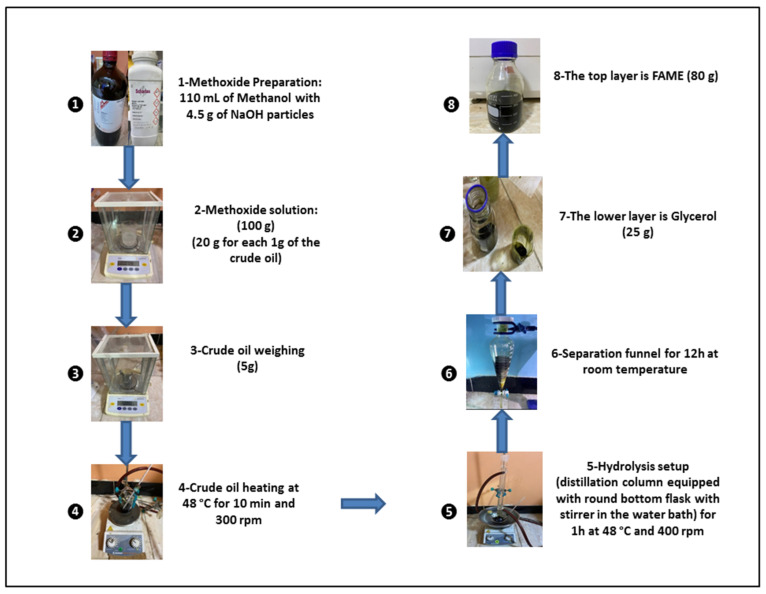
Experimental images for the hydrolysis procedure for the crude oil of *Chlorella vulgaris* microalgae.

**Figure 4 molecules-27-08018-f004:**
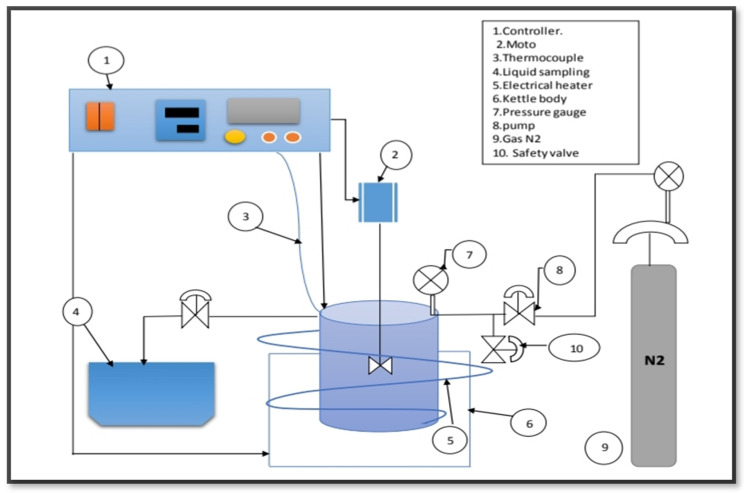
Schematic diagram of the used batch reactor.

**Figure 5 molecules-27-08018-f005:**
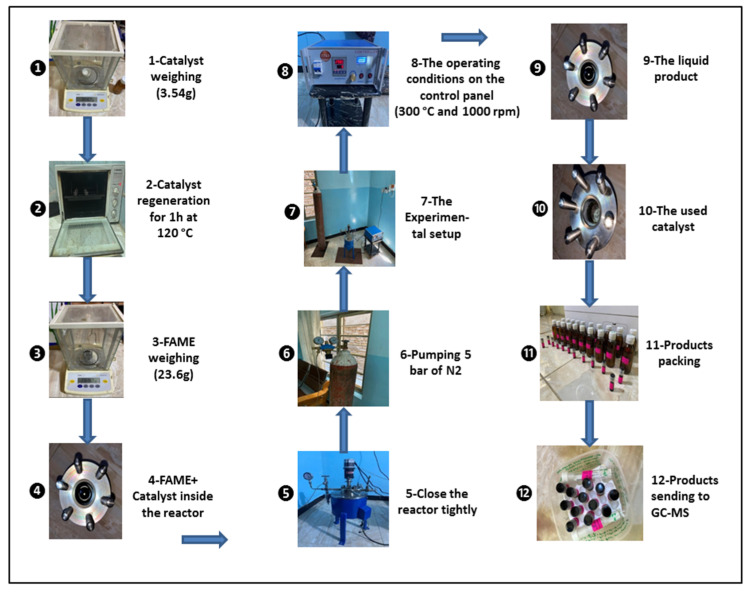
Experimental images for the procedure of conducting the catalytic deoxygenation reactions for the hydrolyzed oil of *Chlorella vulgaris* microalgae.

**Figure 6 molecules-27-08018-f006:**
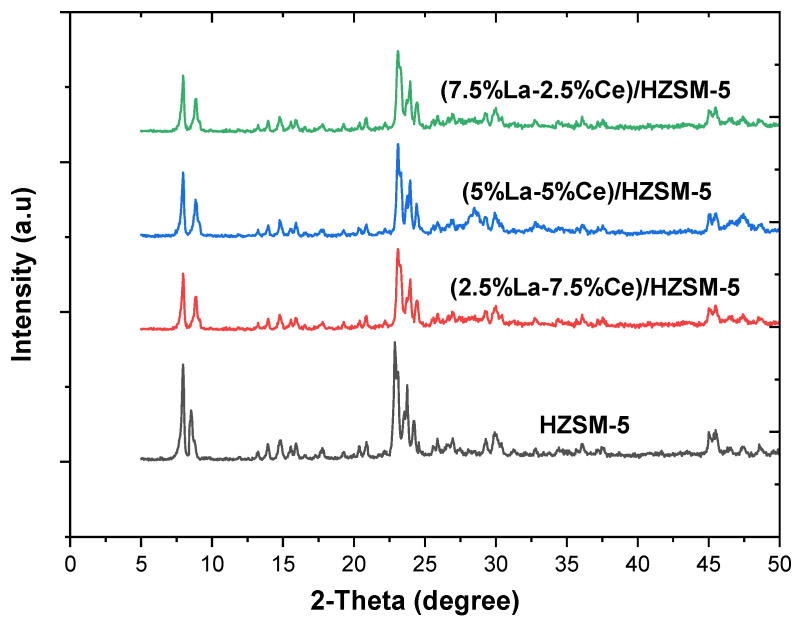
XRD patterns for parent HZSM-5 and Lanthanum-Cerium-modified zeolite catalysts.

**Figure 7 molecules-27-08018-f007:**
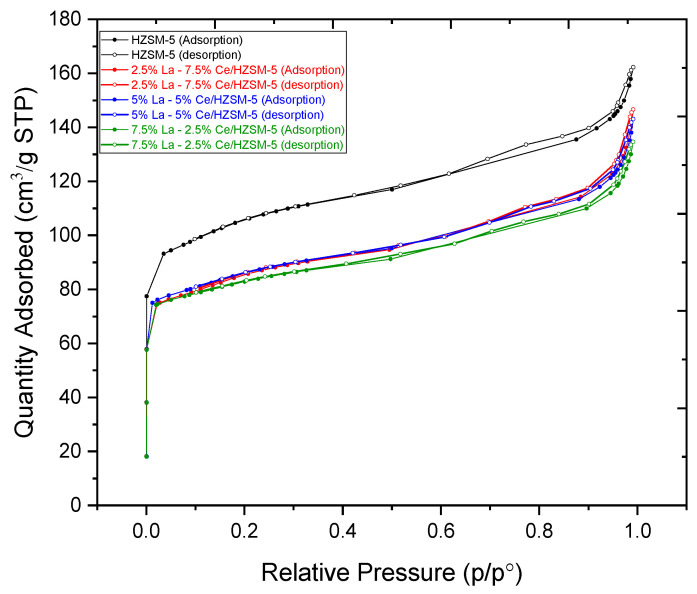
N_2_ adsorption-desorption isotherms of the parent HZSM-5 and Lanthanum-Cerium-modified HZSM-5 with different loading weight percentages.

**Figure 8 molecules-27-08018-f008:**
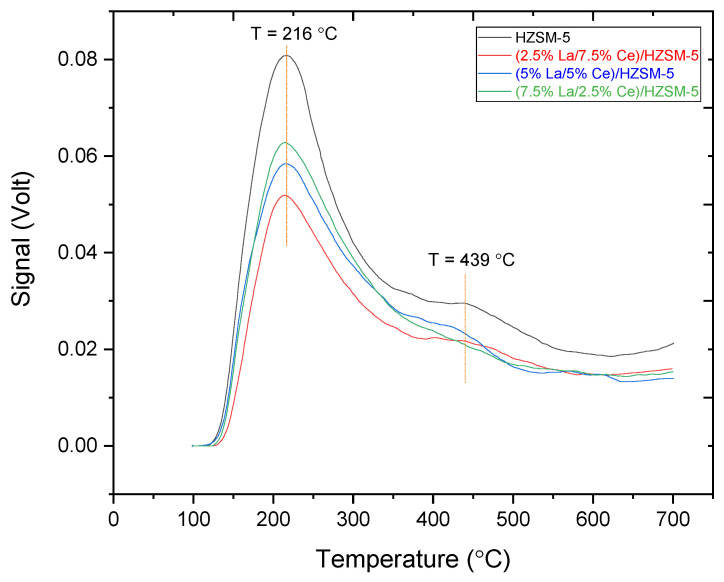
NH_3_-TPD profiles of the parent HZSM-5 and Lanthanum-Cerium modified catalysts: (2.5%La-7.5%Ce)/HZSM-5, (5%La-5%Ce)/HZSM-5, and (7.5%La-2.5%Ce)/HZSM-5.

**Figure 9 molecules-27-08018-f009:**
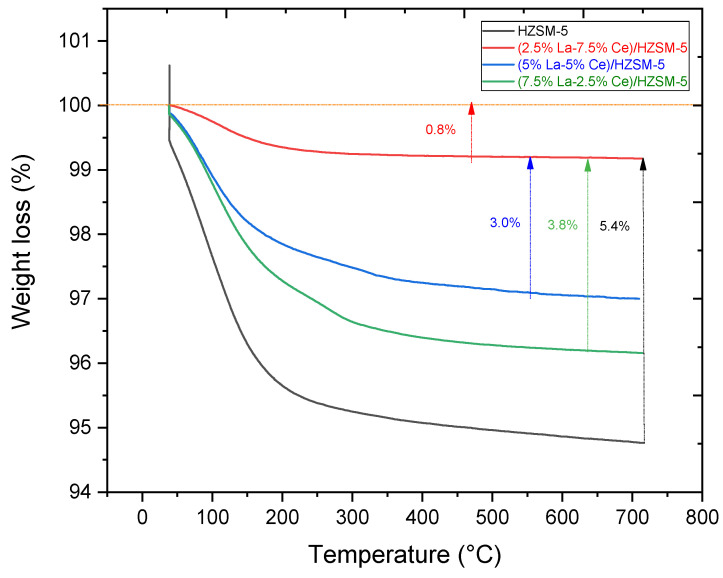
TGA of the fresh parent HZSM-5 and fresh Lanthanum-Cerium-modified HZSM-5 with different loading weight percentages.

**Figure 10 molecules-27-08018-f010:**
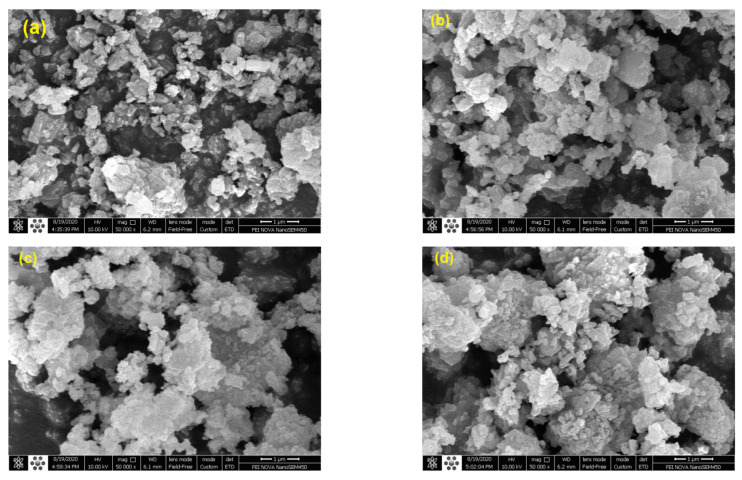
SEM images of HZSM-5 (**a**), (2.5%La-7.5%Ce)/HZSM-5 (**b**), (5%La-5%Ce)/HZSM-5 (**c**), and (7.5%La-2.5%Ce)/HZSM-5 (**d**).

**Figure 11 molecules-27-08018-f011:**
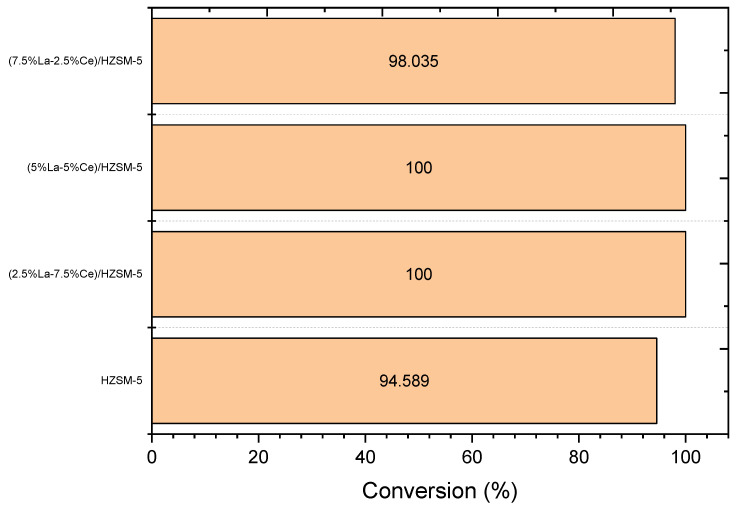
Conversions of the algal HO of the catalytic deoxygenation reactions for the parent HZSM-5, (2.5%La-7.5%Ce)/HZSM-5, (5%La-5%Ce)/HZSM-5, and (7.5%La-2.5%Ce)/HZSM-5 at specified operating conditions (batch reactor, 300 °C, 1000 rpm, 7 bar of N_2_ inert gas (initial pressure), catalyst-to-algal HO ratio = 15% (wt.%), and 6 h).

**Figure 12 molecules-27-08018-f012:**
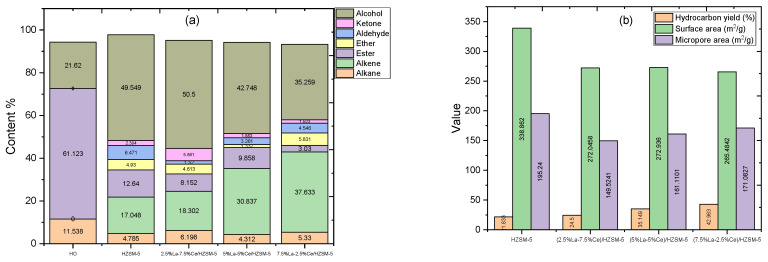
(**a**) The chemical composition groups of the algal HO and the liquid products from the catalytic deoxygenation of the algal HO for the parent HZSM-5 zeolite and Lanthanum-Cerium-modified HZSM-5 zeolite with different loading weight percentages (batch reactor, 300 °C, 1000 rpm, 7 bar N_2_, catalyst-to-HO ratio = 15% (wt.%), and 6 h); (**b**) hydrocarbon yield percentage distribution with the surface area and micropore area of the synthesized catalysts; (**c**) hydrocarbon yield percentage distribution with the acidity of the synthesized catalysts.

**Figure 13 molecules-27-08018-f013:**
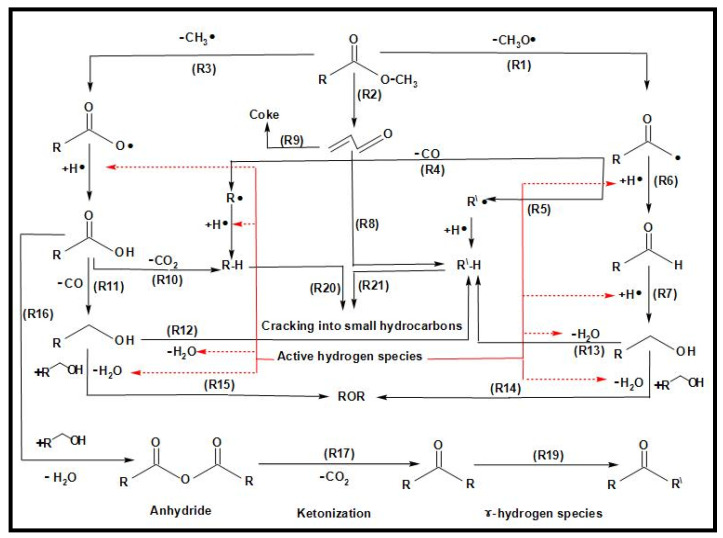
Proposed overall reaction pathways for the catalytic cracking of the algal HO for parent HZSM-5 and Lanthanum-Cerium modified zeolites.

**Figure 14 molecules-27-08018-f014:**
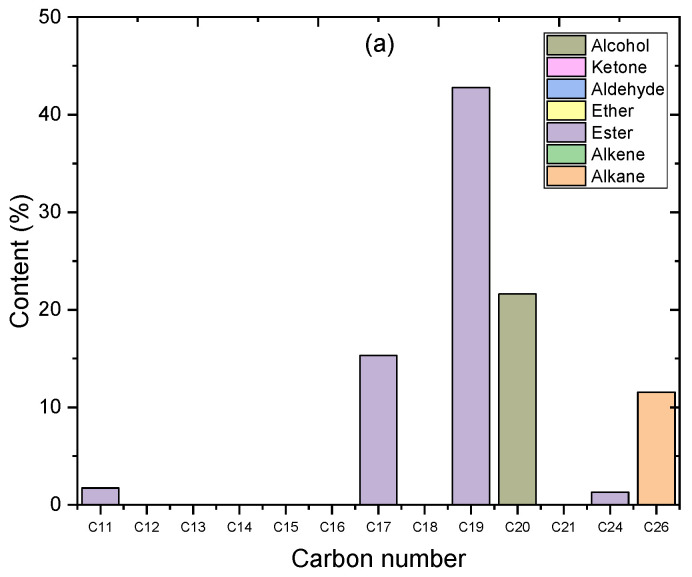
(**a**). Carbon number distribution for the algal HO. (**b**). Carbon number distribution for the products of the catalytic deoxygenation of the algal HO for HZSM-5. (**c**). Carbon number distribution for the products of the catalytic deoxygenation of the algal HO for (2.5%La-7.5%Ce)/HZSM-5. (**d**). Carbon number distribution for the products of the catalytic deoxygenation of the algal HO for (5%La-5%Ce)/HZSM-5. (**e**). Carbon number distribution for the products of the catalytic deoxygenation of the algal HO for (7.5%La-2.5%Ce)/HZSM-5.

**Figure 15 molecules-27-08018-f015:**
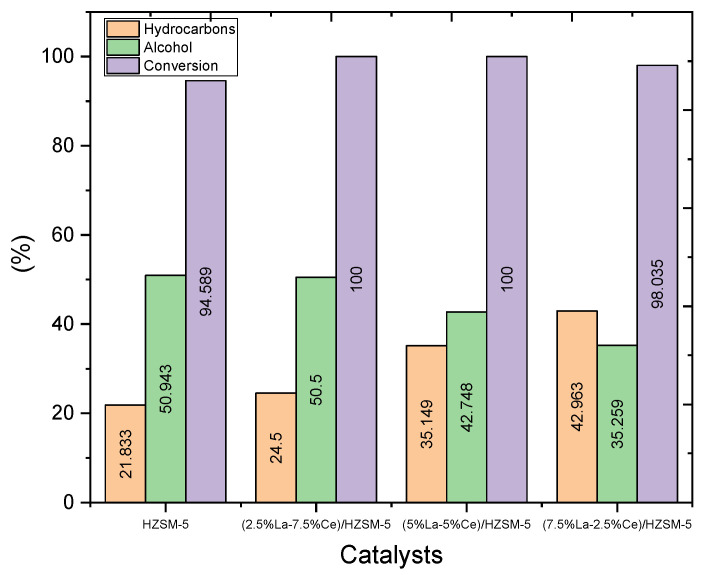
The conversion percentage of the algal HO and the yield percentages of the outstanding chemicals of the hydrocarbons and alcohol from the catalytic deoxygenation of the algal HO for the parent HZSM-5 zeolite and Lanthanum-Cerium-modified HZSM-5 zeolite with different loading weight percentages (batch reactor, 300 °C, 1000 rpm, 7 bar N_2_ initial gas (inert gas), catalyst-to-algal HO ratio = 15% (wt.%), and 6 h).

**Figure 16 molecules-27-08018-f016:**
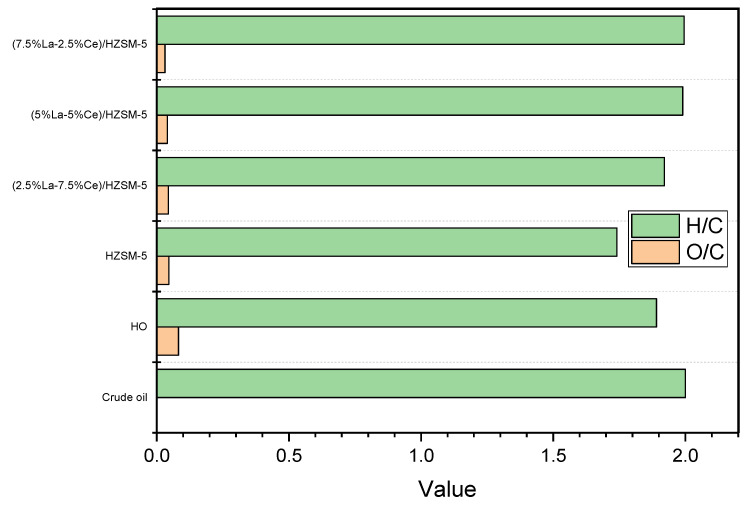
van Krevelen diagram of the liquid products produced by catalytic deoxygenation of the algal HO for HZSM-5 and Lanthanum-Cerium-modified zeolite catalysts.

**Table 1 molecules-27-08018-t001:** The relative crystallinity for parent HZSM-5 and Lanthanum-Cerium-modified zeolite catalysts.

**No.**	**Catalyst Name**	**Relative Crystallinity** **(** **%** **)**
1	HZSM-5	100
2	(2.5%La-7.5%Ce)/HZSM-5	79
3	(5%La-5%Ce)/HZSM-5	79
4	(5%La-5%Ce)/HZSM-5	79

**Table 2 molecules-27-08018-t002:** Texture properties of the parent HZSM-5 and Lanthanum-Cerium-modified HZSM-5 with different loading weight percentages of modified HZSM-5.

No.	Catalyst	S_BET_ (m^2^/g)	**S_micro_ (m^2^/g)**	**S_extern_ (m^2^/g)**	**V_total_ (cm^3^/g)**	**V_micro_ (cm^3^/g)**	Average Particle Size (nm)
1	HZSM-5	338	195	143	0.223	0.100	17
2	(2.5%La-7.5%Ce)/HZSM-5	272	149	122	0.190	0.078	22
3	(5%La-5%Ce)/HZSM-5	272	161	111	0.189	0.084	21
4	(7.5%La-2.5%Ce)/HZSM-5	265	171	94	0.180	0.086	22

S_BET_: BET surface area was calculated by Brumauer–Emmett–Teller (BET) mode. S_micro_: Micropore area was determined from the t-plot micropore area. S_extern_: External surface area was determined from the t-plot area. V_total_: The total pore volumes were obtained from the adsorbed amount at P/P_0_ = 0.95. V_micro_: The micropore volume was measured by the t-plot method.

**Table 3 molecules-27-08018-t003:** NH_3_-TPD properties of HZSM-5, (2.5%La-7.5%Ce)/HZSM-5, (5%La-5%Ce)/HZSM-5, and (7.5%La-2.5%Ce)/HZSM-5.

Catalyst	Low Peak Temperature Point (°C) (Weak Acid Peak)			High Peak Temperature Point (°C) (Strong Acid Peak)			Total Acid Amount (Total NH_3_ Amount mmol/g)
	T (°C)	TCD (V)	NH_3_ Amount (mmol/g)	T (°C)	TCD (V)	NH_3_ Amount (mmol/g)	
**HZSM-5**	216	0.0808	0.526	439	0.0295	0.214	0.74
**(2.5%La-7.5%Ce)/HZSM-5**	214.2	0.0519	0.346	401.4	0.02241	0.178	0.524
**(5%La-5%Ce)/HZSM-5**	216.7	0.05839	0.394	400.1	0.02549	0.190	0.584
**(7.5%La-2.5%Ce)/HZSM-5**	214.2	0.06281	0.418	400	0.02401	0.184	0.602

**Table 4 molecules-27-08018-t004:** Mass balances (wt. %) of the algal HO compound in the feed and the compounds of liquid products for the conversion of HO for HZSM-5, (2.5%La-7.5%Ce)/HZSM-5, (5%La-5%Ce)/HZSM-5, and (7.5%La-2.5%Ce)/HZSM-5.

**Compounds of the Algal HO**	**Molecular Formula**	**Content of the Compound in the Feed ** **(** **HO** **) (** **wt.%** **)**	**Content** **(** **wt.%** **)** **of the Compound in the Liquid Product of the Catalytic Deoxygenation Reactions for the Algal HO as a Function of the Lanthanum-Cerium Loading Percentage on the Parent HZSM-5**			
			**HZSM-5**	(2.5%La-7.5%Ce)/**HZSM-5**	(5%La-5%Ce)/**HZSM-5**	(7.5%La-2.5%Ce)/**HZSM-5**
Hexacosane	C_26_H_54_	11.538	0	0	0	0
6-Octen-1-ol, 3,7-dimethyl-, formate	C_11_H_20_O_2_	1.728	0	0	0	0
9,12,15-Octadecatrienoic acid, methyl ester, (Z,Z,Z)-	C_19_H_32_O_2_	15.084	0	0	0	0
Hexadecanoic acid, methyl ester	C_17_H_34_O_2_	15.325	5.411	0	0	1.965
9,12-Octadecadienoic acid, methyl ester	C_19_H_34_O_2_	27.692	0	0	0	0
Di-n-octyl phthalate	C_24_H_38_O_4_	1.294	0	0	0	0
Phytol	C_20_H_40_O	21.620	40.859	22.38	34.091	26.352
others	-	5.719	0	0	0	0
**Conversion (%) of the HO in the catalytic deoxygenation reactions as a function of the Lanthanum-Cerium loading percentage on the parent HZSM-5**			**94.589**	**100**	**100**	**98.035**

**Table 5 molecules-27-08018-t005:** The main components and content of the hydrolyzed oil (HO) and the products of catalytic deoxygenation for the HO for the parent HZSM-5 zeolite and Lanthanum-Cerium-modified HZSM-5 zeolite with different loading weight percentages (batch reactor, 300 °C, 1000 rpm, 7 bar N_2_ inert gas (initial pressure), catalyst-to-algal HO ratio = 15% (wt.%), and 6 h).

Compound	Molecular Formula	Hydrolyzed oil (HO)	HZSM-5	(2.5%La-7.5%Ce)/**HZSM-5**	(5%La-5%Ce)/**HZSM-5**	(7.5%La-2.5%Ce)/**HZSM-5**
**ALKANE**						
Hexacosane	C_26_H_54_	11.538				
Tetradecane	C_14_H_30_		4.785			
Nonadecane	C_19_H_40_			3.113	4.312	5.330
Bicyclo [3.1.1]heptane, 2,6,6-trimethyl-, (1.alpha.,2.beta.,5.alpha.)	C_10_H_18_			3.085		
**TOTAL ALKANES**		**11.538**	**4.785**	**6.198**	**4.312**	**5.330**
**ALKENS**						
5-Ethyl-1-nonene	C_11_H_22_		17.048	11.592	24.827	30.055
1-Undecene, 8-methyl-	C_12_H_24_			3.339	6.010	7.578
Supraene	C_30_H_50_			3.371		
**TOTAL ALKENS**		**0**	**17.048**	**18.302**	**30.837**	**37.633**
**ESTERS**						
6-Octen-1-ol, 3,7-dimethyl-, formate	C_11_H_20_O_2_	1.728				
9,12,15-Octadecatrienoic acid, methyl ester, (Z,Z,Z)-	C_19_H_32_O_2_	15.084				
Hexadecanoic acid, methyl ester	C_17_H_34_O_2_	15.325	5.411			1.965
Carbonic acid, butyl undec-10-enyl ester	C_16_H_30_O_3_		1.883			
9,12-Octadecadienoic acid, methyl ester	C_19_H_34_O_2_	27.692				
Di-n-octyl phthalate	C_24_H_38_O_4_	1.294				
trans-13-Octadecenoic acid, methyl ester	C_19_H_36_O_2_		5.346			
6-Octadecenoic acid, methyl ester, (Z)-	C_19_H_36_O_2_			2.006		
Cyclopentanetridecanoic acid, methyl ester	C_19_H_36_O_2_			1.063		
Hexadecanoic acid, 2-hydroxy-, methyl ester	C_17_H_34_O_3_			5.083		
Tridecanoic acid, 12-methyl-, methyl ester	C_15_H_30_O_2_				3.184	
11-Octadecenoic acid, methyl ester	C_19_H_36_O_2_				3.847	
Fumaric acid, 2,4-dimethylpent-3-yl tridecyl ester	C_24_H_44_O_4_				1.011	
2,2-Dimethylpropanoic acid, 2,6-dimethylnon-1-en-3-yn-5-yl ester	C_16_H_26_O_2_				1.816	
Butyl 9-tetradecenoate	C_18_H_34_O_2_					1.065
**TOTAL ESTERS**		**61.123**	**12.64**	**8.152**	**9.858**	**3.03**
**ETHERS**						
Disparlure	C_19_H_38_O		1.468			
Tetrahydropyran 12-tetradecyn-1-ol ether	C_19_H_34_O_2_		1.379			
Oxirane, tridecyl-	C_15_H_30_O		2.083			
2-Furanmethanamine, tetrahydro-	C_5_H_11_NO			4.613		
Octadecane, 1-(ethenyloxy)-	C_20_H_40_O				1.332	
2H-Pyran, 2-[(2-furanylmethoxy)methyl]tetrahydro-	C_11_H_16_O_3_					1.504
9-Octadecene, 1-[3-(octadecyloxy)propoxy]-, (Z)-	C_39_H_78_O_2_					4.327
**TOTAL ETHERS**		**0**	**4.93**	**4.613**	**1.332**	**5.831**
**ALDEHYDES**						
Tetradecanal	C_14_H_28_O		6.471	1.52	3.261	
2-Heptadecenal	C_17_H_32_O					4.546
**TOTAL ALDEHYDES**		**0**	**6.471**	**1.52**	**3.261**	**4.546**
**KETONES**						
2-Pentadecanone, 6,10,14-trimethyl	C_18_H_36_O		2.394			
B(9a)-Homo-19-norpregna-9(11),9a-dien-20-one, 3-(dimethylamino)-4,4,14-trimethyl-, (3.beta.,5.alpha.)-	C_26_H_41_NO			5.861		
2(4H)-Benzofuranone, 5,6,7,7a-tetrahydro-4,4,7a-trimethyl-, (R)-	C_11_H_16_O_2_				0.998	
Benz[e]azulene-3,8-dione, 5-[(acetyloxy)methyl]-3a,4,6a,7,9,10,10a,10b-octahydro-3a,10a-dihydroxy-2,10-dimethyl-, (3a.alpha.,6a.alpha.,10.beta.,10a.beta.,10b.beta.)-(+)-	C_19_H_24_O_6_				0.885	
2(1H)-Naphthalenone, octahydro-4a, 5-dimethyl-3-(1-methylethyl)-, (3. alpha.,4a.beta.,5.beta.,8a.alpha.)	C_14_H_24_O					1.623
**TOTAL KETONES**		**0**	**2.394**	**5.861**	**1.883**	**1.623**
**ALCOHOLS**						
1-Dodecanol, 3,7,11-trimethyl-	C_15_H_32_O		8.690			
Phytol	C_20_H_40_O	21.620	40.859	22.38	34.091	26.352
2-Methyl-Z,Z-3,13-octadecadienol	C_19_H_36_O			2.113		
1,22-Docosanediol	C_22_H_46_O_2_			1.415		
2-Methyl-Z,Z-3,13-octadecadienol	C_19_H_36_O			10.186		
Cyclododecanol, 1-ethenyl-	C_14_H_26_O			14.406		
3,7,11,15-Tetramethyl-2-hexadecen-1-ol	C_20_H_40_O				7.341	5.698
1,1’-Bicyclopentyl-1,1’-diol	C_10_H_18_O_2_				1.316	
trans-2-Dodecen-1-ol	C_12_H_24_O					1.947
1-Decanol, 2-hexyl-	C_16_H_34_O					1.262
**TOTAL ALCOHLS**		**21.620**	**49.549**	**50.5**	**42.748**	**35.259**
**TOTAL Areas (%)**		**94.281**	**97.817**	**95.146**	**94.231**	**93.252**
**Others Areas (%) = 100 Total Areas (%)**		**5.719**	**2.183**	**4.854**	**5.769**	**6.748**

**Table 6 molecules-27-08018-t006:** Product yield percentages of outstanding hydrocarbon (alkanes and alkenes) and alcohol compounds from catalytic deoxygenation of the algal HO for the parent and Lanthanum-Cerium-modified zeolites at 300 °C for 6 h under initial N_2_ pressure of 7 bar, at 1000 rpm and with 23.6 g of algal HO/3.54 g of the catalyst in the batch reactor.

Hydrocarbon Compound	Molecular Formula	Hydrolyzed Oil (HO)	HZSM-5	(2.5%La-7.5%Ce)/HZSM-5	(5%La-5%Ce)/HZSM-5	(7.5%La-2.5%Ce)/HZSM-5
Hexacosane	C_26_H_54_	11.538				
Tetradecane	C_14_H_30_		4.785			
Nonadecane	C_19_H_40_				4.312	5.330
Tridecane, 7-hexyl-	C_19_H_40_			3.113		
Bicyclo [3.1.1]heptane, 2,6,6-trimethyl-, (1.alpha.,2.beta.,5.alpha.)	C_10_H_18_			3.085		
5-Ethyl-1-nonene	C_11_H_22_		17.048	11.592	24.827	30.055
1-Undecene, 8-methyl-	C_12_H_24_			3.339	6.010	7.578
Supraene	C_30_H_50_			3.371		
**The total yield of the hydrocarbons compounds**		**11.538**	**21.833**	**24.5**	**35.149**	**42.963**
**Alcohol Compound**	**Molecular Formula**	**Hydrolyzed Oil (HO)**	**HZSM-5**	**(2.5%La-7.5%Ce)/HZSM-5**	**(5%La-5%Ce)/HZSM-5**	**(7.5%La-2.5%Ce)/HZSM-5**
1-Dodecanol, 3,7,11-trimethyl-	C_15_H_32_O		8.69			
Phytol	C_20_H_40_O	21.620	40.859	22.38	34.091	26.352
2-Methyl-Z,Z-3,13-octadecadienol	C_19_H_36_O			2.113		
1,22-Docosanediol	C_22_H_46_O_2_			1.415		
2-Methyl-Z,Z-3,13-octadecadienol	C_19_H_36_O			10.186		
Cyclododecanol, 1-ethenyl-	C_14_H_26_O			14.406		
3,7,11,15-Tetramethyl-2-hexadecen-1-ol	C_20_H_40_O				7.341	5.698
1,1’-Bicyclopentyl-1,1’-diol	C_10_H_18_O_2_				1.316	
trans-2-Dodecen-1-ol	C_12_H_24_O					1.947
1-Decanol, 2-hexyl-	C_16_H_34_O					1.262
**The total yield of the alcohols compounds**		**21.620**	**49.549**	**50.5**	**42.748**	**35.259**

**Table 7 molecules-27-08018-t007:** Hydrocarbon production via catalytic deoxygenation with various catalyst types in references and catalytic deoxygenation investigated in this study.

**Reactant**	**Catalyst**	**Reactant/Catalyst Ratio**	**Reactant/Solvent**	**Reactor Type**	**Pressure (Bar), Gas**	**Temperature (°C)**	**Time (h)**	**Conversion (%)**	**Observations**	Ref.
palm kernel oil	HBeta zeolite	10/1.5	-	B.R	10 bar H_2_	350	5	-	The total yield of hydrocarbons = 82 ± 3%	[13]
Hydrolyzed palm kernel oil	HBeta zeolite	10/1.5	-	B.R	10 bar H_2_	350	5	-	The total yield of hydrocarbons = 24 ± 9%	[13]
Olein oil	HBeta zeolite	10/1.5	-	B.R	10 bar H_2_	350	5	-	The total yield of hydrocarbons = 43 ± 3%	[13]
Hydrolyzed olein oil	HBeta zeolite	10/1.5	-	B.R	10 bar H_2_	350	5	-	The total yield of hydrocarbons 98 ± 4%	[13]
Hydrolyzed Macauba oil	HBeta zeolite	10/1	-	B.R	10 bar H_2_	350	5	-	The total yield of hydrocarbons = 30%	[13]
Hydrolyzed castor oil	5% Pd/C	1/0.1	1 g Hydrolyzed castor oil/30 mL n-hexane	B.R	25 bar H_2_	310	7	-	The total yield of hydrocarbons = 57%	[14]
Hydrolyzed castor oil	5% Pd/C	1/0.1	1 g Hydrolyzed castor oil/30 mL n-dodecane	B.R	25 bar H_2_	310	7	-	The total yield of hydrocarbons = 39.6%	[14]
Hydrolyzed castor oil	5% Pd/C	1/0.1	1 g Hydrolyzed castor oil/30 mL n-hexane	B.R	25 bar H_2_	300	7	-	The total yield of hydrocarbons = 40%	[14]
Hydrolyzed castor oil	5% Pd/C	1/0.1	1 g Hydrolyzed castor oil/30 mL n-hexane	B.R	25 bar H_2_	340	7	-	The total yield of hydrocarbons ~96%	[14]
Stearic acid	10%Ni/HZSM-5 (Si/Al = 40)	1/0.2	1 g stearic acid/100 mL dodecane	B.R	40 bar H_2_	260	8		Total selectivity of hydrocarbons ~56%	[15]
Microalgae oil	10%Ni/HBeta (Si/Al = 180)	1/0.2	1 g Microalgae oil/100 mL dodecane	B.R	40 bar H_2_	260	6		The total yield of hydrocarbons = 70%	[15]
Crude oilof Microalgae	10%Ni/ZrO_2_	1/0.5	-	B.R	40 bar H_2_	270	6	-	The total yield of hydrocarbons = 72%	[16]
Crude oilof Microalgae	10%Ni/ZrO_2_	1/0.5	-	B.R	40 bar H_2_	270	4	-	The total yield of hydrocarbons = 61%	[16]
Palmitic acid	Ni/LY char	1/1	1 g Palmtic acid/10 g hexane	B.R	30 bar H_2_	300	5	31.41	The total yield of hydrocarbons = 12.75%	[17]
Palmitic acid	Ni/LY char	1/1	1 g Palmtic acid/10 g acetone	B.R	30 bar H_2_	300	5	67	The total yield of hydrocarbons = 12.49%	[17]
Methyl oleate	5% Pd/C	0.83 mol/L/1 g of catalyst	-	Semi-batch	15 bar H_2_	300	6	96	Total selectivity of hydrocarbons = 29%	[18]
Methyl oleate	5% Pd/C	0.83 mol/L/1 g of catalyst	-	Semi-batch	15 bar Ar	300	6	44	Total selectivity of hydrocarbons = 17%	[18]
Soybean oil	20%Ni/Al_2_O_3_	50/0.55	-	B.R	7 bar N_2_	350	4	74	Total yield of hydrocarbons = 79.5%	[26]
Stearic acid	Pd/Al_2_O_3_	1	-	B.R	7 bar N_2_	350	6	43	Total selectivity of hydrocarbons = 35%	[27]
Cellulose, and glycerol	HZSM-5(Si/Al = 36)	cellulose: glycerol: catalyst = 1:0.05:0.004	100 g of n-heptane	B.R	-	350	0.5	-	The total yield of hydrocarbons = 21%	[28]
Cellulose, and glycerol	5%Fe/HZSM-5 (Si/Al = 36)	cellulose: glycerol: catalyst = 1:0.05:0.004	100 g of n-heptane	B.R	-	350	0.5	-	Total yield of hydrocarbons = 38%	[28]
Lauric acid	5% Pd/C	1/0.1	1 g of acid/100 mL of hexadecane	S.B.R	20 bar Ar	300	6	-	Total yield of hydrocarbons = 38	[29]
Lauric acid	5% Pd/C	1/0.1	1 g of acid/100 mL of hexadecane	S.B.R	20 bar Ar	300	3	-	Total yield of hydrocarbons = 28	[29]
Algal HO	HZSM-5(Si/Al = 30)	1 g of algal HO/0.15 g of the catalyst	-	B.R	7 bar N_2_	300	6	94.589	The total yield hydrocarbons = 21.833%	This study
Algal HO	(2.5%La-7.5%Ce)/HZSM-5 (Si/Al = 30)	1 g of algal HO/0.15 g of the catalyst	-	B.R	7 bar N_2_	300	6	100	The total yield of hydrocarbons = 24.5%	This study
Algal HO	5%La-5%Ce)/HZSM-5 (Si/Al = 30)	1 g of algal HO/0.15 g of the catalyst	-	B.R	7 bar N_2_	300	6	100	The total yield of hydrocarbons = 35.149%	This study
Algal HO	7.5%La-2.5%Ce)/HZSM-5 (Si/Al = 30)	1 g of algal HO/0.15 g of the catalyst	-	B.R	7 bar N_2_	300	6	98.035	The total yield of hydrocarbons = 42.963%	This study

**Table 8 molecules-27-08018-t008:** The degree of deoxygenation, elemental composition, higher heating value, H/C and O/C atomic ratios and the algal HO, and the liquid products of the catalytic deoxygenation for the parent HZSM-5 and Lanthanum-Cerium-modified zeolite catalysts.

NO.	Liquid Type	Element (%)			HHV (MJ/Kg)	H/C (Mole Ratio)	O/C (Mole Ratio)	DOD%
		C	H	O				
1	Hydrolyzed oil (HO)	78.916	12.432	8.651	32.377	1.890	0.082	n.a.
2	Liquid product for HZSM-5	82.905	12.026	5.068	33.230	1.740	0.045	44.235
3	Liquid product for (2.5% La-7.5%Ce)/HZSM-5	82.072	13.136	4.790	33.738	1.920	0.043	46.755
4	Liquid product for (5% La-5%Ce)/HZSM-5	82.024	13.602	4.373	34.036	1.989	0.039	51.368
5	Liquid product for (7.5% La-2.5%Ce)/HZSM-5	82.796	13.764	3.439	34.362	1.994	0.031	62.109
6	Crude oil [118]	83–86	11–14	˂1	44	1.5–2.0	~0	n.a.

n.a.: not applicable.

## Data Availability

Not applicable.

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
