# Peer review of "Bimetallic Lanthanum-Cerium-Loaded HZSM-5 Composite for Catalytic Deoxygenation of Microalgae-Hydrolyzed Oil into Green Hydrocarbon Fuels"

_molecules, 2022, doi:10.3390/molecules27228018_

Round 1
Reviewer 1 Report
This manuscript by Prof. Alias and Prof. Tahir et al. describes the preparation of La-Ce modified HZSM-5 composite materials and their applications toward the catalytic conversion of microalgae hydrolyzed oil to hydrocarbon products. The author has prepared 3 kinds of modified HZSM-5 materials with different weight ratio of La/Ce loading. The conversion reaction by the catalysts is examined and the products are characterized by GC-MS. The results indicate that the (7.5% La-2.5% Ce)/HZSM-5 catalyst exhibits the best catalytic performance: the hydrocarbon yield of 42.96%, HHV of 34.36 MJ/kg, DOD% of 62.19%, which are better than the parent HZSM-5 one. The results are exciting as the La-Ce HZSM-5 materials have not been used to catalyze the conversion reaction of microalgae hydrolyzed oil. The results can provide some valuable information for further investigations. There are some comments to the manuscript prior to acceptance by the Journal.
(1) Throughout the manuscript, the number of acidic sites are mentioned as the important factors that govern the formation of coke. The loading of La and Ce is confirmed to lower the number of acidic sites. In the main text, Lewis and Bronsted acid sites are frequently discussed to have different influences on the catalytic conversion and the coke formation. In some paragraphs, these 2 different types of acidic sites are separately discussed. In the other paragraphs, the author mixes both for the discussion. It is recommended to clearly distinguish one kind of the acid site from the other. In addition, their impacts on the catalytic selectivity and the coke formation shall be clearly made.
(2) It is still unclear that the loading of La-Ce causes the changes of the Lewis vs Bronsted acid sites.
(3) When “7.5% La-2.5% Ce” is claimed, is the actual elemental weight analysis performed?
(4) The influence of the La-Ce loading on the acid site due to Al not part of the framework or inside zeolitic framework is not clearly presented. Is there any data to support the statement?
(5) Regarding the production distribution, the data presentation is too trivial. It is better to discuss the importance of the La-Ce loading on the changes of the product distribution in the macroscopic view as well as the specific product, instead of stating the data of the products. Any insightful conclusion on conversion selectivity on the basis of the varied loading ratio?
(6) The data presentation can be re-organized to more readable fashion.
(7) Several important factors like solvent used, working temperature, and H2 initial pressure have major influences on the product selectivity. They should be studied.
(8) In Fig. 12c, for the modified HZSM-5 catalysts, the increase the acidity, the increase the hydrocarbon yield. This result is inconsistent with the observation that the parent HZSM-5 has the highest acidity and the lowest hydrocarbon yield.
(9) The scope of content of this manuscript is better suited for the following journal: Catalysts.
Author Response
The manuscript has been revised carefully by considering all the comments and suggestions as reported in the attached file.

Reviewer 2 Report
This manuscript described the La-Ce doped HZSM-5 for deoxygenation of oil. It seems that authors tried to explain all procedure of experiments and discuss their results. However several parts of this article should be revised.
Page 4, line 180: Only four catalysts were prepared and compared performance and characteristics. It would be more systematic study if only La- and Ce-doped HZSM-5 catalysts (two more samples) were prepared. Then it should be discussed or described in manuscript why each sample were not prepared.
Page 7, line 254: The loading amount of La and Ce needs to be checked by using ICP or XRF study. If there is the result about catalyst component, it needs to be added in manuscript.
Page 8, line 304: Why gas phase product was not measured? It should be more described in manuscript and explain the expected gases.
Page 9, line 322: Usually GC-MS is known as a qualitative characterization. Thus calibration depending on the concentration of product seems to be pre-conducted.
Page 11, Figure 6: Error on X-axis
Page 11, line 376: Too many decimal on BET surface area, pore volume and pore diameter
Page 13, line 424: Error on X-axis
Page 15, line 490: Author described that TG result indicates that La- and Ce-doped catalyst leads to less coke deposition. I don’t think that there is no evidence of this discussion. It should be more described or deleted.
Page 17, line 525: The catalyst amount on performance test doesn’t described either experimental or discussion part. Plus, each catalyst (parent and doped catalysts) showed difference conversion, however its difference is too low to discuss the enhancement of performance. I think it might be in error range.
Page 18, Table 4: Although 2.5% La-7.5% Ce/HZSM-5 and 5%La-5%Ce/HZSM-5 showed the same conversion, the content of phytol was different. It should be discussed in manuscript.
Page 23, Figure 12: Hydrocarbon yield is not correlated to catalyst acidity and increases depending on La content. What about add more La over HZSM-5? And what is each role of La and Ce promoter over HZSM-5 in reaction?
Page 26, Figure 13: Author said that gas phase product was not measured in this study. However the mechanism of catalytic cracking includes gas phase products such as CO and CO2. Then this mechanism could make reader confused. I think that author should notice this kinds of limitation.
Author Response
The manuscript has been carefully revised by considered all the comments and suggestions as attached.

Round 2
Reviewer 1 Report
The author has made corresponding changes based on the reviewer comments.
It is, however, recommended to cite the author's recent paper on La-HZSM-5 in various loading %, and the Ce work. Citing the work can help ones to clarify any confusion concerning the impact of Ln-loading.
Author Response
The response to reviewer comments has been summarized in a Table as attached.

Reviewer 2 Report
I think that manuscript has revised to some extent. In response file, author describes that there are two other articles related to this work. I can't know which article would be uploaded first. If it is possible, it would be better to add two articles as reference for readers.
Author Response

(The authors gave the same response as above.)

Round 3
Reviewer 1 Report
The reviewer recommends the acceptance of the revision as the author has made the corresponding changes.
Author Response
-